# A bio-inspired visuotactile neuron for multisensory integration

Muhtasim Ul Karim Sadaf[1,5], Najam U Sakib[1,5], Andrew Pannone[1], Harikrishnan Ravichandran[1] & Saptarshi Das [1,2,3,4] ✉

Multisensory integration is a salient feature of the brain which enables better and faster responses in comparison to unisensory integration, especially when the unisensory cues are weak. Specialized neurons that receive convergent input from two or more sensory modalities are responsible for such multisensory integration. Solid-state devices that can emulate the response of these multisensory neurons can advance neuromorphic computing and bridge the gap between artificial and natural intelligence. Here, we introduce an artificial visuotactile neuron based on the integration of a photosensitive monolayer $MoS_2$ memtransistor and a triboelectric tactile sensor which minutely captures the three essential features of multisensory integration, namely, super-additive response, inverse effectiveness effect, and temporal congruency. We have also realized a circuit which can encode visuotactile information into digital spiking events, with probability of spiking determined by the strength of the visual and tactile cues. We believe that our comprehensive demonstration of bio-inspired and multisensory visuotactile neuron and spike encoding circuitry will advance the field of neuromorphic computing, which has thus far primarily focused on unisensory intelligence and information processing.

Relying on visual senses for navigation in complete darkness is not useful; instead, tactile senses can be more effective. While a hard touch can reveal more information about an object or an obstacle owing to large neural responses, a soft touch may be inadequate in evoking neural feedback. However, hard touches can also lead to undesired consequences such as damage to an object, e.g., during locomotion inside a dark room with valuable artwork or injury to the body due to the presence of dangerous entities. In such situations, even a short-lived flash of light can significantly enhance the chance of successful locomotion. This is because visual memory can subsequently influence and aid the tactile responses for navigation. This would not be possible if our visual and tactile cortex were to respond to their respective unimodal cues alone. Integration of cross-modal cues is, therefore, one of the essential features of how the brain functions.

In the brain, each sense functions optimally under different circumstances, but collectively they can enhance the likelihood of detecting and identifying objects and events. Commonly, it is believed that there are dedicated areas in the brain, such as the visual, auditory, somatosensory, gustatory, and olfactory cortices, that process sensory input from one modality, whereas cross-modal integrations occur in higher cortical areas. However, recent findings show that multisensory integration can take place in primary sensory areas via specialized neurons that receive convergent inputs from two or more sensory modalities[1]. For example, S1 neurons found in the primary somatosensory cortex of trained monkeys respond to visual and auditory stimuli in addition to somatosensory inputs[2,3]. Similarly, A1 neurons in the auditory cortex respond to both auditory and somatosensory cues[4]. The advantage of multisensory integration is that the multisensory response

[1]Engineering Science and Mechanics, Penn State University, University Park, PA 16802, USA. [2]Electrical Engineering, Penn State University, University Park, PA 16802, USA. [3]Materials Science and Engineering, Penn State University, University Park, PA 16802, USA. [4]Materials Research Institute, Penn State University, University Park, PA 16802, USA. [5]These authors contributed equally: Muhtasim Ul Karim Sadaf, Najam U Sakib. ✉e-mail: sud70@psu.edu

is super-additive, i.e., it not only exceeds the individual unisensory responses but also their arithmetic sum.

Another key feature of multisensory integration is that multisensory enhancement is typically inversely related to the strength of the individual cues that are being combined[5]. This is referred to as the inverse effectiveness effect and makes intuitive sense, as highly salient unimodal stimuli will evoke vigorous responses in corresponding unisensory neurons, which can be easily detected. In contrast, weak cues are comparatively difficult to detect via unisensory neurons; in such cases, multisensory integration can substantially enhance neural activity and positively impact animal behavior by increasing the speed and likelihood of detecting and locating an event[6–8]. In other words, multisensory amplification is the greatest when responses evoked by individual stimuli are the weakest. Finally, multisensory integration demonstrates temporal congruency, i.e., the magnitude of the integrated response is sensitive to the temporal correlation between the responses that are initiated by each sensory input[9]. In other words, the response is maximal when the peak periods of activity coincide.

Examples of multisensory information processing are abundant in nature. Dolphins, for instance, combine auditory cues derived from echoes with their visual system, enabling them to develop a comprehensive understanding of objects, distances, and shapes present in their environment. Honeybees communicate the whereabouts of food sources to their hive mates through intricate dances called "waggle dances." These dances incorporate visual cues, such as the angle and duration of the waggle, along with odor cues obtained from the nectar, effectively conveying information about the food source's distance and direction. Electric fish integrate sensory inputs from their electric sense, vision, and mechanosensation to form a comprehensive perception of their surroundings. While multisensory integration has been widely studied in neuroscience, particularly in the context of cognition and behavior, its benefits are yet to be fully utilized in the fields of robotics, artificial intelligence, and neuromorphic computing.

Note that there are some recent demonstrations of neuromorphic devices that can respond to more than one external stimulus. For example, Liu et al.[10] demonstrated a stretchable and photoresponsive nanowire transistor that can perceive both tactile and visual information, You et al.[11] demonstrated visuotactile integration using piezoresistors and $MoS_2$ field-effect transistors (FETs), Jiang et al.[12] used commercial sensors and spike encoding circuits to encode bimodal motion signals such as acceleration and angular speed and subsequently integrated the two using a dual gated $MoS_2$ FET, Wang et al.[13] demonstrated gesture recognition by integrating visual data with somatosensory data from stretchable sensors, Yu et al.[14] realized a mechano-optic artificial synapse based on a graphene/$MoS_2$ heterostructure and an integrated triboelectric nanogenerator, and Sun et al.[15] reported an artificial reflex arc that senses/processes visual and tactile information using a self-powered optoelectronic perovskite (PSK) artificial synapse and controls artificial muscular actions in response to environmental stimuli. The visual and somatosensory information was also encoded as impulse spikes. Similarly, Chen et al.[16] reported a $CsPbBr_3$/$TiO_2$-based floating-gate transistor that can respond to both light and temperature, and Han et al.[17] proposed a fingerprint recognition system based on a single-transistor neuron (1T-neuron) that can integrate visual and thermal stimuli. Finally, Yuan et al.[18] demonstrated $VO_2$-based artificial neurons that can encode illuminance, temperature, pressure, and curvature signals into spikes, and Liu et al.[19] reported an artificial autonomic nervous system to emulate the joint action of sympathetic and parasympathetic nerves on organs to control the contraction and relaxation of artificial pupils and visually simulate normal and abnormal heart rates. However, none of the neuromorphic devices mentioned above embrace the true characteristic features of multisensory integration, i.e., super-additivity, inverse effectiveness effect, and temporal congruency. Furthermore, except for the study by Sun et al.[15], none of the above studies

demonstrated spike encoding of multisensory information. Here, we introduce a neuromorphic visuotactile device by integrating a triboelectric tactile sensor with a photosensitive monolayer $MoS_2$ memtransistor that can mimic the characteristic features and functionalities of a multisensory neuron (MN). A benchmarking table highlighting the advances made in this work over previous demonstrations on multisensory integration is shown in Supplementary Information 1.

## Bio-inspired MN

Figure 1a shows a schematic representation of the multisensory integration of visual and tactile information within the biological nervous system, and Fig. 1b shows our bio-inspired visuotactile MN comprising a tactile sensor connected to the gate terminal of a monolayer $MoS_2$ photo-memtransistor as well as the associated spike encoding circuit. The tactile sensor exploits the triboelectric effect to encode touch stimuli into electrical impulses, which are subsequently transcribed into source-to-drain output current spikes ($I_{DS}$) in the $MoS_2$ photo-memtransistor. Similarly, visual stimuli are encoded into threshold voltage shifts by exploiting the photogating effect in monolayer $MoS_2$ photo-memtransistors. The encoding circuit is also built using $MoS_2$ memtransistors. The entire experimental setup is shown in Supplementary Fig. 1. Figure 1c summarizes the three characteristic features of multisensory integration, i.e., super-additive response to cross-modal cues, inverse effective effect, and temporal congruency. In other words, our artificial visuotactile neuron and encoding circuitry can mimic the essential attributes of multisensory integration.

In this study, we have used monolayer $MoS_2$ grown via metal–organic chemical vapor deposition (MOCVD) on an epitaxial sapphire substrate at 1000 °C. Details on material synthesis, film transfer, and device fabrication can be found in the Methods section and in our previous works[20–27]. Preliminary material characterization, which includes Raman and photoluminescence spectroscopic analysis of monolayer $MoS_2$, and electrical characterization, which includes the transfer and output characteristics of $MoS_2$ photo-memtransistors measured in the dark, are shown in Supplementary Fig. 2a–d, respectively. All $MoS_2$ devices used in this study have channel length $L_{CH} = 1\,\mu m$ and channel width $W_{CH} = 5\,\mu m$, as shown using the plan-view optical micrograph in Supplementary Fig. 3. For the cross-sectional transmission electron microscopy (TEM) image and energy-dispersive X-ray spectroscopy (EDS) demonstrating the elemental distribution, please refer to our recent study in which we employed an identical device stack[28]. Note that while any photo-memtransistor can be used for this demonstration, the use of monolayer $MoS_2$ is motivated by the fact that beyond visual[27] and tactile[29] sensations, $MoS_2$-based transistors can be used as chemical sensors[30], gas sensors[31], temperature sensors[32], and acoustic sensors[33], which greatly expands the scope for multisensory integration to gustatory, olfactory, thermal, and auditory sensations as well. In addition, $MoS_2$-based devices have enabled various neuromorphic and bio-inspired applications through the integration of sensing, computing, and storage capabilities[34–40]. Finally, $MoS_2$ is among the most mature two-dimensional (2D) materials and can be grown at the wafer scale using chemical vapor deposition techniques[20]; at the same time, aggressively-scaled $MoS_2$-based transistors[41] with near Ohmic contacts[42] have achieved high performance that meets the IRDS standards for advanced technology nodes[43–45].

## Unisensory tactile and visual response of the MN

First, we study the response of our MN to tactile and visual stimuli alone. The tactile response is obtained using the triboelectric effect, where electrical impulses are generated due to charge transfer when two dissimilar materials come into contact. For our demonstration, the tactile sensor is composed of a stack of commercially available Kapton and aluminum foil separated by an air gap. PDMS stamps with different surface areas were prepared to serve as the touch stimuli (see

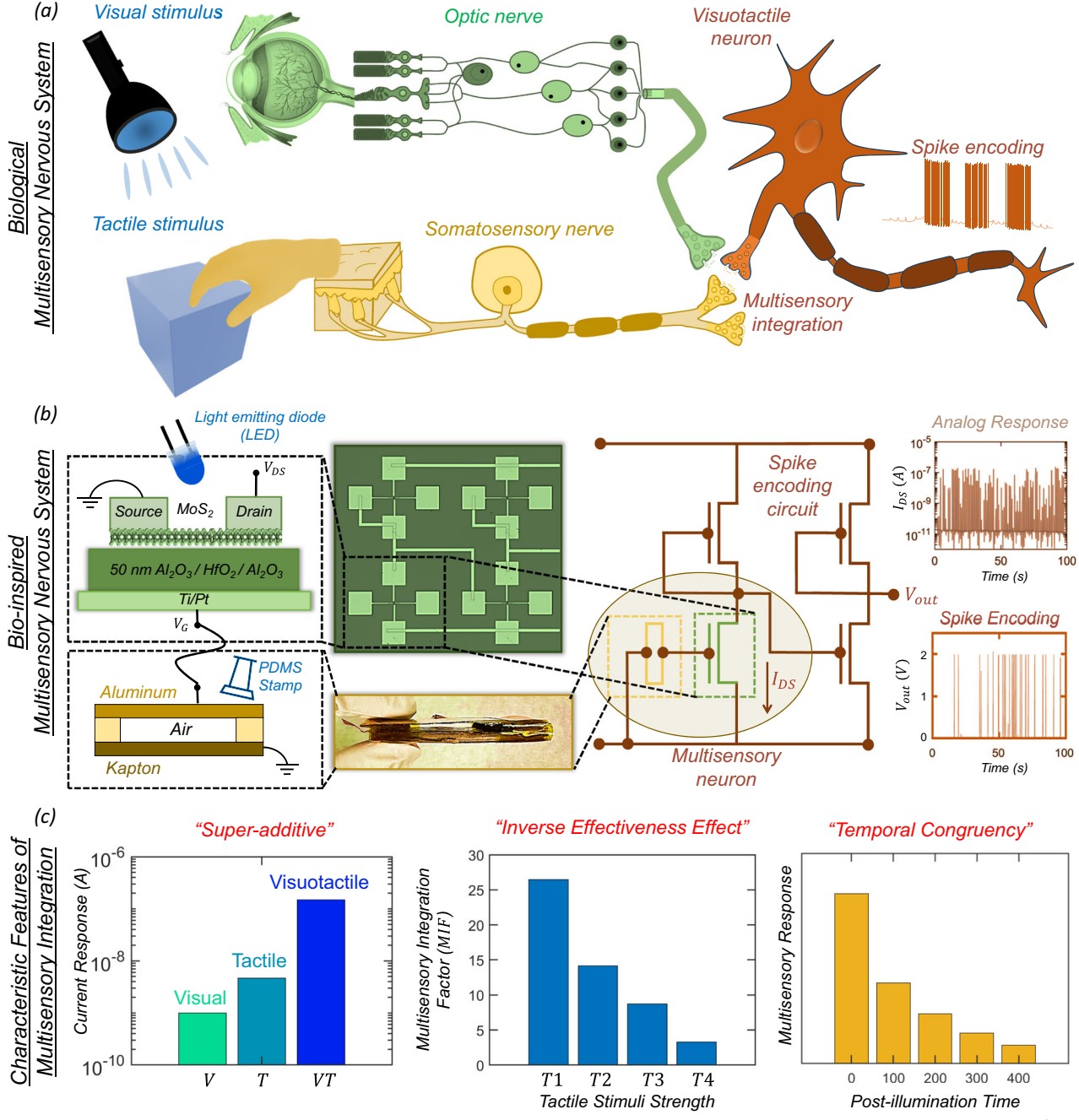

**Fig. 1 | Multisensory integration. a** Schematic representation of multisensory integration of visual and tactile information within the biological nervous system. **b** A bio-inspired visuotactile multisensory neuron (MN) comprising a triboelectric tactile sensor connected to the gate terminal of a monolayer $MoS_2$ photo-memtransistor along with the associated spike encoding circuit. Electrical impulses generated by the tactile sensor are transcribed into source-to-drain output current spikes ($I_{DS}$) by the $MoS_2$ photo-memtransistor. Similarly, visual stimuli are encoded into threshold voltage shifts by exploiting the photogating effect in monolayer $MoS_2$ photo-memtransistors. The encoding circuit is also built using $MoS_2$ memtransistors to convert analog $I_{DS}$ spikes to digital voltage spikes ($V_{out}$). **c** The three characteristic features of multisensory integration, i.e., super-additive response to cross-modal cues, inverse effective effect, and temporal congruency, are demonstrated by our bio-inspired visuotactile MN.

Supplementary Fig. 4). Note that the magnitude of the electrical impulse generated by our triboelectric tactile sensor is directly proportional to the surface charge, which is strongly dependent on the surface contact area ($T$). Since the tactile sensor is connected to the gate terminal of the $MoS_2$ photo-memtransistor, the electrical impulse ($V_{spike}$) generated by the touch gets encoded as source-to-drain current ($I_{DS}$) spikes at the output of the MN. Figure 2a shows the MN response ($I_{DS}$ spikes) for different tactile stimuli under dark conditions ($I_{LED} = 0$ A) with the touch inputs having contact surface areas of

$T1 = 25$ mm², $T2 = 49$ mm², $T3 = 100$ mm², and $T4 = 400$ mm², respectively. A source-to-drain supply voltage ($V_{DS}$) of 1 V was applied across the $MoS_2$ photo-memtransistor. Supplementary Fig. 5a shows the histogram of $I_{DS}$ spikes. All touch inputs given to the tactile sensor started from a height of approximately 1 cm with a tapping frequency of ~1 Hz. For any given touch stimulus, there is some inherent variation in the magnitude of $I_{DS}$ spikes, which is typical of the triboelectric effect. However, with the increasing size of the touch input, the magnitude of the $I_{DS}$ spikes also increases, indicating that more tactile information is

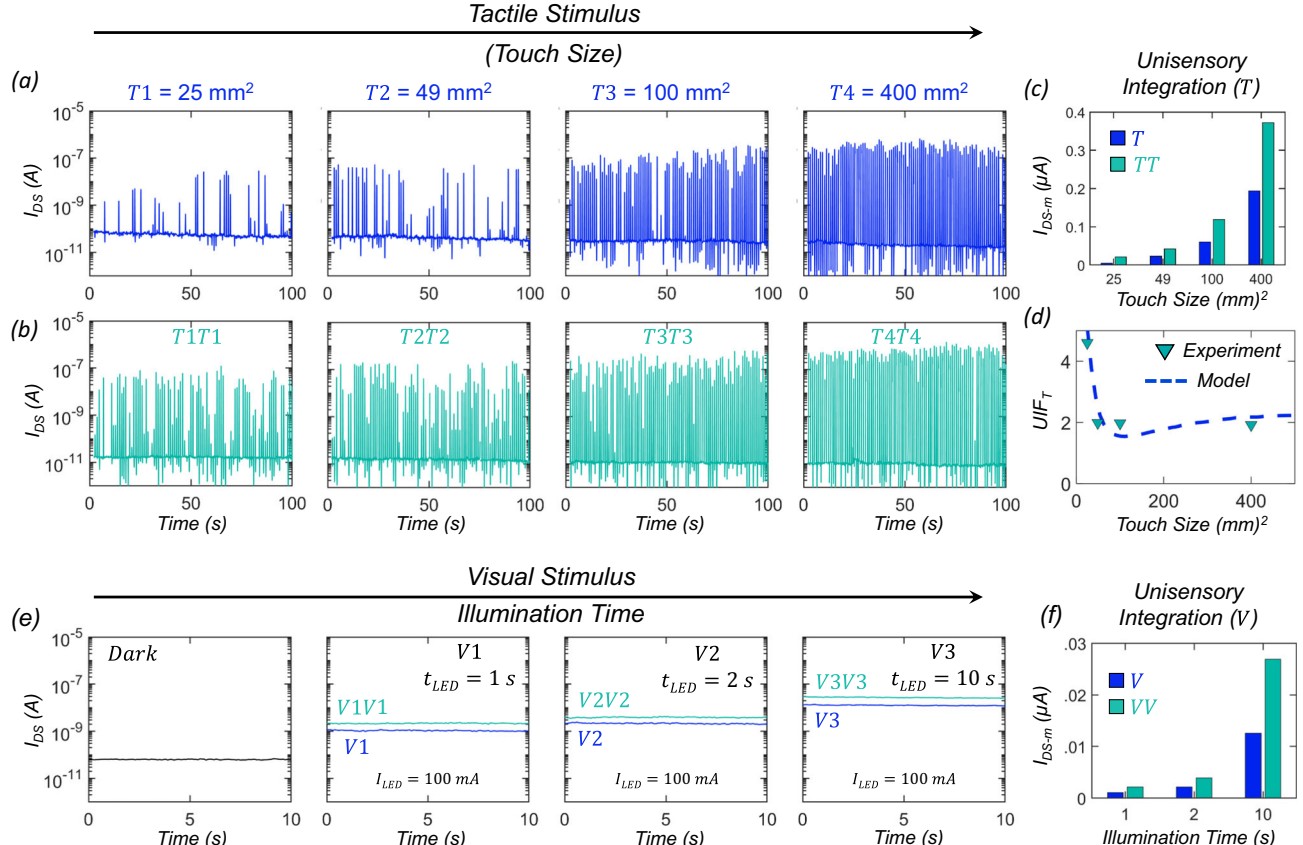

**Fig. 2 | Tactile and visual response of MN and unisensory integration. a** Tactile response from MN, i.e., source-to-drain current ($I_{DS}$) spikes obtained from the MoS$_2$ photo-memtransistor under dark condition with increasing strength of touch size, $T1 = 25$ mm$^2$, $T2 = 49$ mm$^2$, $T3 = 100$ mm$^2$, and $T4 = 400$ mm$^2$, respectively. **b** Response of the MN to two identical touch inputs applied simultaneously, with each input having the same area as mentioned above. For better visualization of the tactile simulation, we have included Supplementary Videos 1 and 2 for single and dual touches, respectively. **c** Bar plot of extracted median values for the source-to-drain current spike magnitude ($I_{DS-m}$) as a function of the strength of the touch size for both single ($T$) and dual ($TT$) touches in linear scale. **d** Unisensory integration factor for tactile stimuli (UIF$_T$), defined as the ratio of $I_{DS-m}$ for dual to single touch responses from the MN, as a function of strength of $T$. **e** Pre-illumination dark current and post-illumination persistent photocurrent response of the MN to different visual stimuli ($V$) obtained from a light emitting diode (LED) with constant input current ($I_{LED} = 100$ mA) and varying illumination time ($t_{LED}$); $V1 = 1$ s, $V2 = 2$ s, and $V3 = 10$ s, respectively. Both single ($V$) and dual ($VV$) illuminations were used to study the unisensory visual integration of the MN. **f** Bar plot of $I_{DS-m}$ as a function of the strength of $V$ for both single and dual illuminations.

required to navigate in the dark. Furthermore, the additive tactile response of the MN was investigated by applying two identical touch inputs simultaneously, with each input having the same area as mentioned above. For better visualization of the tactile simulation, we have included Supplementary Videos 1 and 2 for single and dual touches, respectively. Figure 2b shows the corresponding $I_{DS}$ spikes and Supplementary Fig. 5b shows the corresponding $I_{DS}$ histograms. Figure 2c and Supplementary Fig. 5c show the bar plot of median spike magnitude ($I_{DS-m}$) as a function of the strength of the tactile stimuli, i.e., touch contact area for both single and dual touches, in linear and logarithmic scale, respectively. Clearly, the MN's response is enhanced for double touch inputs ($TT$) compared to a single touch ($T$) input, demonstrating the unisensory integration capability of our MN. Figure 2d shows the unisensory integration factor for tactile stimuli (UIF$_T$), which we define as the ratio of $I_{DS-m}$ for dual to single-touch responses from the MN, as a function of strength of $T$. Note that the unisensory integration is super-additive for the weakest tactile stimulus ($T1$), as UIF ≈ 4.5, whereas, for the strongest tactile stimulus ($T4$), the unisensory integration becomes near additive with UIF ≈ 2.

To explain the unisensory integration response of our bio-inspired MN to tactile stimuli, we have developed an empirical model. We have used the virtual source (VS) model to describe the MoS$_2$ photo-memtransistor using parameters extracted from the experimental transfer characteristics[34,39,46]. In the VS model, both

the subthreshold and the above threshold characteristics of the MoS$_2$ photo-memtransistor are captured through a single semi-empirical relationship described in Eq. 1.

$$I_{DS} = \frac{V_{DS}}{R_{CH}}; R_{CH} = \frac{L_{CH}}{W_{CH}\mu_N Q_{CH}}; Q_{CH} = C_G m \frac{k_B T_a}{q} \log\left[1 + \exp\left(\frac{V_{spike} - V_{TH}}{mk_B T_a/q}\right)\right]$$

(1)

In Eq. 1, $R_{CH}$ is the channel resistance, $L_{CH} = 1$ μm is the channel length, $W_{CH} = 5$ μm is the channel width, $\mu_N$ is the carrier mobility for electrons in MoS$_2$, $Q_{CH}$ is the inversion charge, $C_G$ is the gate capacitance, $V_{spike}$ is the applied gate voltage, $V_{TH}$ is the threshold voltage, $m$ is the band movement factor, $k_B$ is the Boltzmann constant, $T_a$ is the temperature, and $q$ is the electron charge. Note that $\mu_N$ can be extracted from the peak transconductance and was found to be ~8 cm$^2$/V·s. Similarly, $V_{TH}$ was extracted using the iso-current method at 10 nA and was found to be ~0.85 V, and $m$ was extracted from the subthreshold slope (SS = $mk_B T_a \ln 10$) and was found to be ~4.5. Supplementary Fig. 6 shows the VS model fitting of the experimental transfer characteristics. Using this VS model, $I_{DS}$ spikes were mapped to their corresponding $V_{spike}$ values. Supplementary Fig. 7a, b shows the histograms for $V_{spike}$ corresponding to single and dual touches of various strengths. The distributions for $V_{spike}$ were described using Gaussian functions with $\mu_T$ and $\sigma_T$ as the mean and standard deviation,

respectively. Note that the strength of the tactile stimulus ($T$) is captured through $\mu_T$, whereas the uncertainty associated with any triboelectric response is captured through $\sigma_T$. Supplementary Fig. 7c, d, respectively, show $\mu_T$ and $\sigma_T/\mu_T$ as a function of $T$. As expected, $\mu_T$ increases with increasing strength of $T$, which we model using the empirical relationship described in Eq. 2a with $\mu_{01} = 0.75$ V, $T_{01} = 25$ mm$^2$, $\mu_{02} = 5$ V, and $T_{02} = 10000$ mm$^2$ as the fitting parameters; $\sigma_T/\mu_T = 0.25$ was assumed constant. The tactile response was subsequently modeled using Eq. 2b.

$$\mu_T = \mu_{01}\left[1 - \exp\left(-\frac{T}{T_{01}}\right)\right] + \mu_{02}\left[1 - \exp\left(-\frac{T}{T_{02}}\right)\right] \quad (2a)$$

$$V_{\text{spike},T} = \text{rand}(\text{Gaussian}, \mu_T, \sigma_T) \quad (2b)$$

The super-additive response of the MN to weaker tactile stimuli can be explained from the fact that when $V_{\text{spike}} < V_{\text{TH}}$, the magnitude of $I_{DS}$ spikes increases exponentially with the strength of $T$, whereas for $V_{\text{spike}} > V_{\text{TH}}$, the MN operates in the linear regime, leading to additive unisensory integration. Figure 2d shows the model-derived UIF, which captures the experimental findings.

Next, we evaluate the unisensory visual response of our artificial MN. The visual response is obtained by exposing the MN to illumination pulses of constant amplitude ($I_{\text{LED}} = 100$ mA) of varying durations ($t_{\text{LED}}$) from a light-emitting diode (LED). During illumination, photocarriers generated in the monolayer MoS$_2$ channel are trapped at the channel/dielectric interface, which leads to a negative shift in the transfer characteristics of the photo-memtransistor that persists even beyond the illumination. Supplementary Fig. 8a shows the transfer characteristics of the MoS$_2$ photo-memtransistor after being exposed to different visual stimuli ($V$), i.e., $V1 = 1$ s, $V2 = 2$ s, and $V3 = 10$ s, and Supplementary Fig. 8b shows the extracted $V_{\text{TH}}$ as a function of $V$. The phenomenon of a persistent shift in the $V_{\text{TH}}$ is known as the photogating effect and is exploited in many neuromorphic devices and vision sensors[22,27,47–52] (See Supplementary Information 2 for more discussion on the photogating effect). For our demonstration, this emulates the role of visual memory in enhancing the tactile response through multisensory integration. Figure 2e shows the pre-illumination dark current and post-illumination persistent photocurrent response of the MN for different $V$. As expected, with increasing strength of $V$, the MN's response ($I_{DS}$) increases monotonically. Figure 2e also shows the response of the MN when two identical visual stimuli ($VV$) are applied. Figure 2f shows the bar plot for $I_{DS-m}$ as a function of the strength of $V$ for both single and dual illuminations. Supplementary Fig. 9 shows the unisensory integration factor for visual stimuli (UIF$_V$), which we define as the ratio of the MN's response to dual and single illuminations, as a function of strength of $V$. UIF$_V$ ranges from ~1.5 to 2, which confirms the sub-additive/additive nature of visual cues and highlights the unisensory integration capability of our artificial MN to visual stimuli.

## Multisensory visuotactile integration

In this section, we will evaluate the response of the MN to cross-modal cues and compare the results with corresponding unisensory responses. Figure 3a shows the $I_{DS}$ spikes obtained from the MN for different combinations of tactile ($T$) and visual ($V$) cues, and Fig. 3b shows a bar plot of extracted $I_{DS-m}$ as a function of $T$ and $V$ (see Supplementary Fig. 10 for the $I_{DS}$ histograms). As expected, the response of the MN monotonically increases with increasing strength of $T$ for any given $V$. However, the response of the MN also shows a monotonic decrease with increasing strength of $V$. This is the so-called inverse effectiveness effect. The physical origin of this effect lies in the screening of the triboelectric gate voltage, obtained from the touch stimuli, by the trapped charges at the interface induced by the visual stimuli. With increasing strength of the visual stimuli, more photo-generated

carriers become trapped at the interface, leading to more screening of the triboelectric voltage. Interestingly, this effect resonates remarkably well with its biological counterpart, i.e., a clear visual memory naturally diminishes the sensitivity to tactile stimuli. Our empirical model can capture this effect by introducing a visual-memory-induced screening factor ($\alpha_V$) for $V_{\text{spike}}$. The MN's response to cross-modal visual and tactile stimuli can be described by Eqs. 3a–3c.

$$I_{DS} = \frac{W_{\text{CH}}}{L_{\text{CH}}}C_G\mu_N m\frac{k_B T_a}{q}\log\left[1 + \exp\left(\frac{\alpha_V V_{\text{spike},T} - V_{\text{TH},V}}{mk_B T_a/q}\right)\right] \quad (3a)$$

$$V_{\text{TH},V} = V_{\text{TH},0} + V_0\exp\left(-\frac{t_{\text{LED}}}{\tau_{\text{LED}}}\right); V_{\text{TH},0} = 0.2V, V_0 = 0.66V, \tau_{\text{LED}} = 4.1s \quad (3b)$$

$$\alpha_V = \left(\frac{t_{\text{LED}}}{\tau_0}\right)^\gamma; \gamma = -0.6, \tau_0 = 2.2s \quad (3c)$$

Supplementary Fig. 11a, b, respectively, show the experimentally obtained and model-fitted $V_{\text{TH},V}$ and $\alpha_V$. Figure 3c shows the bar plot for $I_{DS-m}$ obtained using the empirical model as a function of $T$ and $V$, which clearly exhibits the inverse effectiveness effect. Next, we assess the benefits of multisensory integration over unisensory responses. Figure 3d shows the comparison of $I_{DS-m}$ obtained from the MN in the presence of multimodal ($VT$) and corresponding unimodal cues. Clearly, the multisensory response exceeds the unisensory responses, as well as their sums, irrespective of the strength of $T$ and $V$. In order to evaluate the effectiveness of multisensory integration, we define a quantity called the multisensory integration factor (MIF) as the ratio of the multisensory response ($I_{DS-m,VT}$) to the sum of individual unisensory responses given by Eq. 4.

$$\text{MIF} = \frac{I_{DS-m,VT}}{I_{DS-m,V} + I_{DS-m,T}} \quad (4)$$

Figure 3e shows the MIF as a function of various $T$ and $V$ stimuli. Note that MIF » 1 for all combinations of $T$ and $V$, thus confirming the super-additive nature of multisensory integration by our artificial MN. We found that the MIF can be as high as ~26.4 when the illumination period is the shortest ($V1 = 1$ s), and the touch area is the smallest ($T1 = 25$ mm$^2$); conversely, the MIF reduces to ~3.3 when the illumination period is the longest ($V3 = 10$ s), and the touch area is the largest ($T4 = 400$ mm$^2$). In other words, MIF decreases monotonically as the strength of the individual cues increases. This makes intuitive sense; offering the greatest multisensory enhancement for the weakest cues can be critical for the survival of the species, whereas diminishing the response when the cues are stronger ensures that the nervous system is not overwhelmed with the multisensory response, highlighting the importance of the inverse effectiveness effect. The inverse effectiveness effect found in the experimentally extracted MIF can also be obtained from the empirical model, as shown in Supplementary Fig. 12.

Next, we compare the effectiveness of multisensory integration against unisensory integration. Figure 4a shows the comparison of $I_{DS-m}$ obtained through multisensory integration ($VT$) against unisensory integration for different strengths of visual ($VV$) and tactile ($TT$) cues. Clearly, the response due to multisensory integration exceeds the response obtained through unisensory integration, irrespective of the strengths of the individual sensory cues. To assess the effectiveness of multisensory integration over unisensory integration, we define MIF$_T$ and MIF$_V$ as the ratio of $I_{DS-m}$ obtained through multisensory integration ($VT$) to unisensory integration, i.e., $TT$ and $VV$, respectively. Figure 4b, c shows the MIF$_T$ and MIF$_V$, respectively. Interestingly, both MIF$_T$ and MIF$_V$ exceed 1 irrespective of the strength

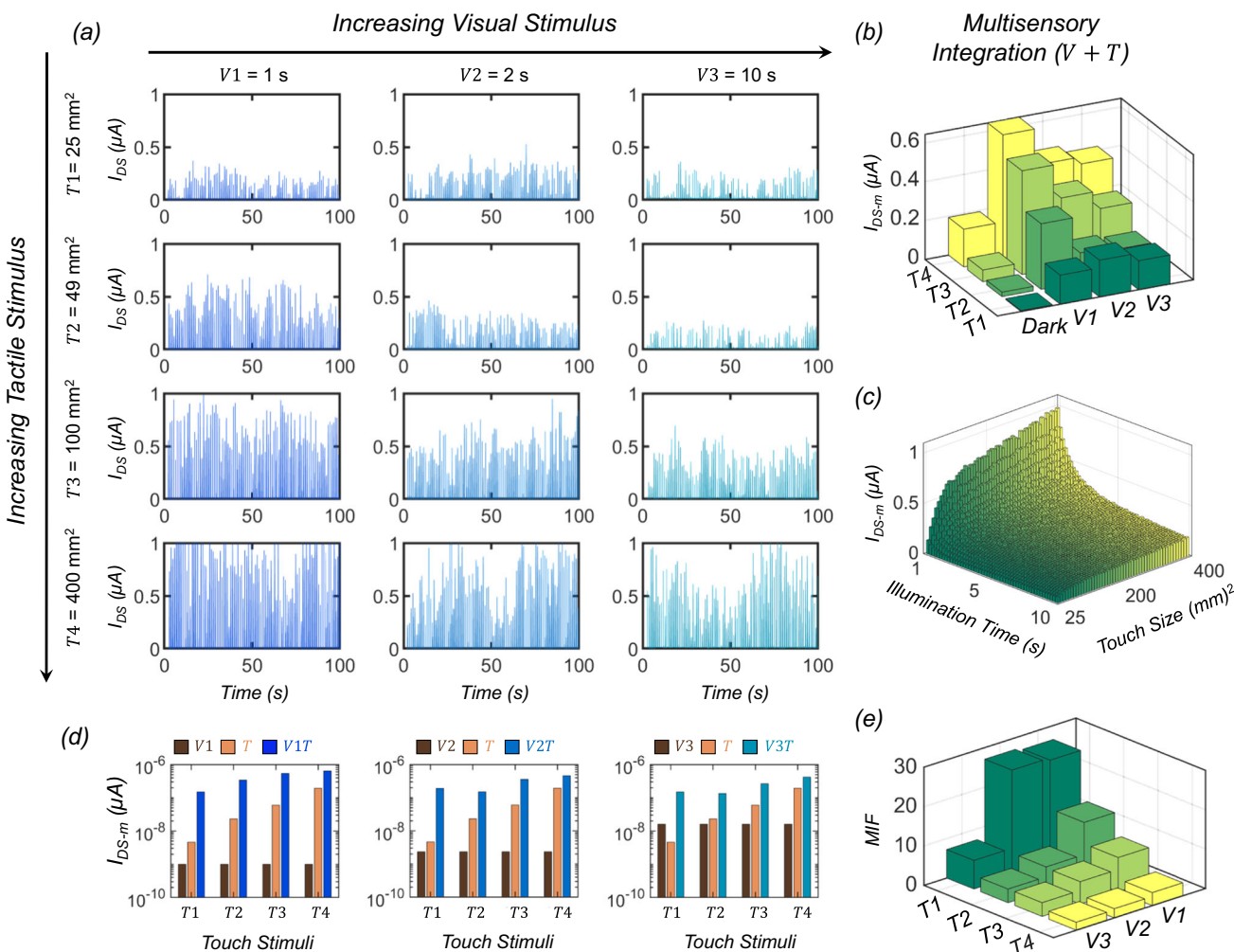

**Fig. 3 | Observation of inverse effectiveness effect and super-additive response from the visuotactile integration by the MN. a** $I_{DS}$ spikes obtained from the MN for different combinations of tactile ($T$) and visual ($V$) cues. **b** Bar plots of extracted $I_{DS-m}$ as a function of $T$ and $V$. While the response of the MN increases with increasing strength of $T$ for any given $V$, as expected, a monotonic decrease in the response of MN with increasing strength of $V$ confirms the inverse effectiveness effect. **c** Results obtained for $I_{DS-m}$ from an empirical model developed for describing the response of the MN to tactile and visual stimuli also exhibit the inverse effectiveness effect. **d** Comparison of $I_{DS-m}$ obtained from the MN in the presence of multimodal ($VT$) and corresponding unimodal cues. Each graph represents the results corresponding to different $V$ and each group of bars within a graph represents results for different $T$; each bar within a group represents $I_{DS-m}$ for $V$, $T$, and $VT$ from left to right. **e** Multisensory integration factor (MIF), defined as the ratio of $I_{DS-m}$ for the multisensory response to the sum of $I_{DS-m}$ for the individual unisensory responses, as a function of $T$ and $V$. Note that MIF >> 1 for all combinations of $T$ and $V$, confirming the super-additive nature of multisensory integration by our artificial MN.

of $T$ and $V$ stimuli, which reinforces the super-additive nature of multisensory integration. For example, $\text{MIF}_T$ can be as high as ~7 when $T$ and $V$ cues are both weak and decrease to ~1.1 with a stronger tactile input. In other words, $\text{MIF}_T$ also demonstrates the inverse effectiveness effect with the strength of the tactile stimulus. Similarly, $\text{MIF}_V$ can be as high as ~66 when both $T$ and $V$ cues are weak. While $\text{MIF}_V$, as expected, increases with increasing strength of $T$ for any given $V$, it shows a monotonic decrease with increasing strength of $V$ for all $T$. In other words, $\text{MIF}_V$ demonstrates the inverse effectiveness effect with the strength of the visual stimulus.

Next, we investigate the temporal congruency offered by our MN. As mentioned earlier, biological MNs show the highest response when cross-modal cues appear simultaneously, while the response falls off monotonically with increasing lag between the cues. Supplementary Fig. 13 shows the response of the MN to touch stimuli as a function of lag ($\triangle \tau$) between the tactile ($T$) and visual ($V$) stimuli, and Fig. 4d shows the corresponding bar plots of $I_{DS-m}$ as a function of $\triangle \tau$ for different $T$. A monotonic decrease can be observed for $I_{DS-m}$ with increasing $\triangle \tau$, confirming that our artificial MN exhibits temporal

congruency. The physical origin of temporal congruency can be attributed to the fact that the persistent photocurrent in $MoS_2$ photo-memtransistors is a direct consequence of photocarrier trapping at the $MoS_2$/dielectric interface; with time, the detrapping process gradually resets the device back to its pre-illumination conductance state. Figure 4e shows the long-term temporal response of the MN after exposure to visual stimuli. Clearly, the post-illumination $I_{DS-m}$ decreases monotonically. This can be regarded as a gradual loss of visual memory. Naturally, tactile cues that appear long after the visual cues are expected to evoke significantly reduced responses. The detrapping process leading to a monotonic decrease in $I_{DS}$ or, equivalently, a monotonic increase in $V_{TH}$ can be described using an exponential decay function with a time constant, $\tau_{detrap} = 260$ s, given by Eq. 5.

$$I_{DS}(t) = I_{DS,0}\exp\left(-\frac{t}{\tau_{detrap}}\right); I_{DS,0} = 28\,\text{nA} \qquad (5)$$

By combining Eqs. 2, 3, and Eq. 5, the phenomenon of temporal congruency can be captured using an empirical model.

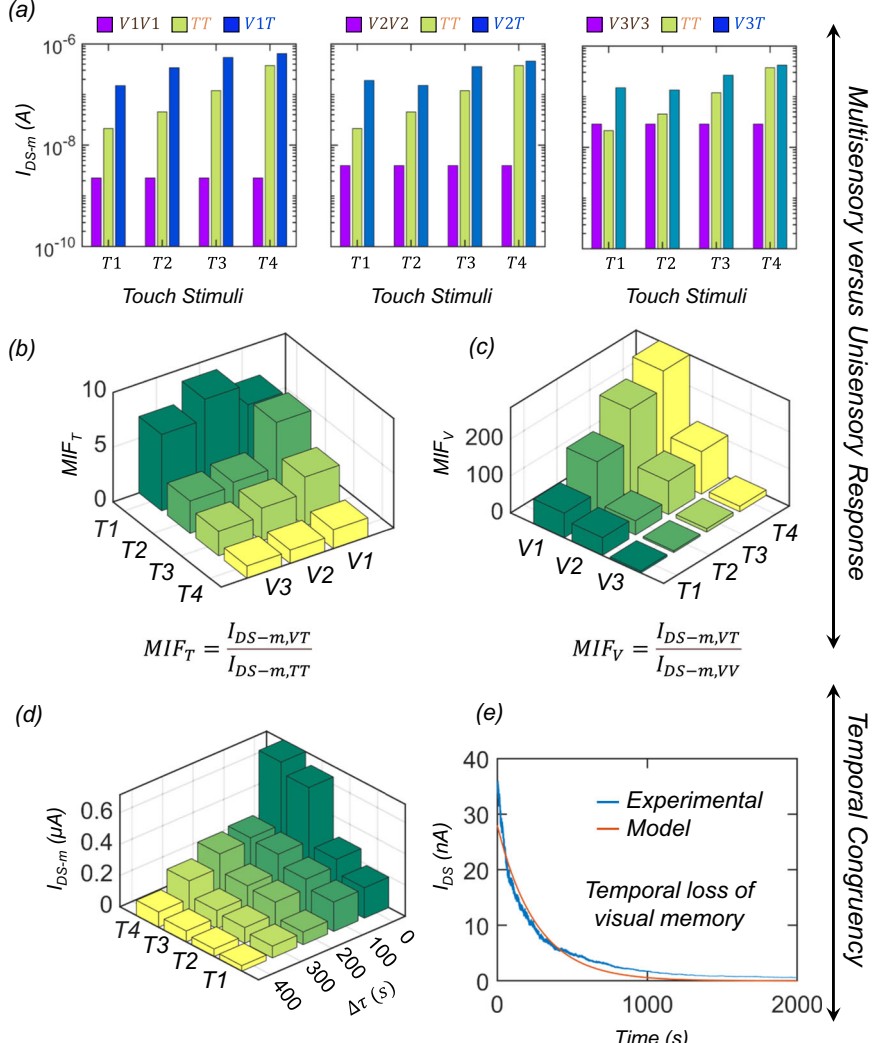

**Fig. 4 | Comparison between unisensory and multisensory integration and demonstration of temporal congruency. a** Bar plots of $I_{DS-m}$ obtained through multisensory integration ($VT$) and unisensory integration for different strengths of visual ($VV$) and tactile ($TT$) cues. Each graph represents the results corresponding to different $V$ and each group of bars within a graph represents results for different $T$; from left to right, each bar within a group represents $I_{DS-m}$ for $VV$, $TT$, and $VT$. Multisensory integration factor for **b** tactile integration (MIF$_T$) and **c** visual integration (MIF$_V$). Both MIF$_T$ and MIF$_V$ exceed 1 irrespective of the strength of $T$ and $V$ stimuli, confirming the advantage of multisensory integration over unisensory integration. **d** Bar plot of $I_{DS-m}$ as a function of temporal lag ($\triangle\tau$) between $V$ and $T$ for different $T$. A monotonic decrease can be observed for $I_{DS-m}$ with increasing $\triangle\tau$, confirming that our artificial MN exhibits temporal congruency. **e** Long-term temporal response of the MN after exposure to visual stimuli. The monotonic decay in persistent photocurrent can be attributed to the gradual detrapping of trapped photocarriers at the dielectric/MoS$_2$ interface.

Note that it is critical to strike a balance between the visual and tactile response for proper functioning of the MoS$_2$ photo-memtransistor-based MN. This can be accomplished by ensuring that $V_{TH-V}$ and $V_{spike,T}$ are of similar magnitudes. To do so, first, the expected strength of the visual ($I_{LED}, t_{LED}, \lambda_{LED}$) and tactile ($T$) stimuli must be determined based on the application requirements and the operating environment. Next, Eq. 1 through Eq. 5 can be self-consistently and iteratively solved to arrive at the required device design dimensions. This is shown schematically in Supplementary Fig. 14. At the same time, it is also important to understand how various device-related parameters influence multisensory integration. Supplementary Fig. 15a–c, respectively, show the dependence of MIF on SS, $\mu_N$, and $V_{TH}$ for the weakest tactile and visual stimuli. Clearly, $\mu_N$ has the least influence on MIF since it is related to the ON-state performance of the photo-memtransistor, whereas visuotactile responses are generated in the OFF-state of the photo-memtransistor. Therefore, as expected, both $V_{TH}$ and SS have a significant impact on MIF, with a more positive $V_{TH}$ and lower magnitude of SS leading to improved

MIF. Note that the dependence of MIF on various device related parameters can become a critical design consideration when an ensemble of multisensory neurons is present. Supplementary Fig. 15d shows the transfer characteristics of 100 multisensory neurons and Supplementary Fig. 15e-g, respectively, show the corresponding neuron-to-neuron variation in SS, $V_{TH}$, and $\mu_N$. The mean values for SS, $V_{TH}$, and $\mu_N$ were found to be 255 mV/decade, 0.42 V, and 10.37 cm$^2$ V$^{-1}$ s$^{-1}$, respectively, with corresponding standard deviation values of 26 mV/decade, 0.15 V, and 6.3 cm$^2$ V$^{-1}$ s$^{-1}$, respectively. Supplementary Fig. 15h shows the projected neuron-to-neuron variation in MIF based on the model discussed earlier. Note that the inherent variation in the tactile response was already built into the model as described in Eq. 2. The mean value for MIF was found to be -18.4 with a standard deviation of -7.3. It is possible to minimize the neuron-to-neuron variation and improve the device performance through further optimization of synthesis, transfer, and cleanliness of the processes associated with device fabrication (see Supplementary Information 3 for more discussion). Finally, the impact of temperature on the

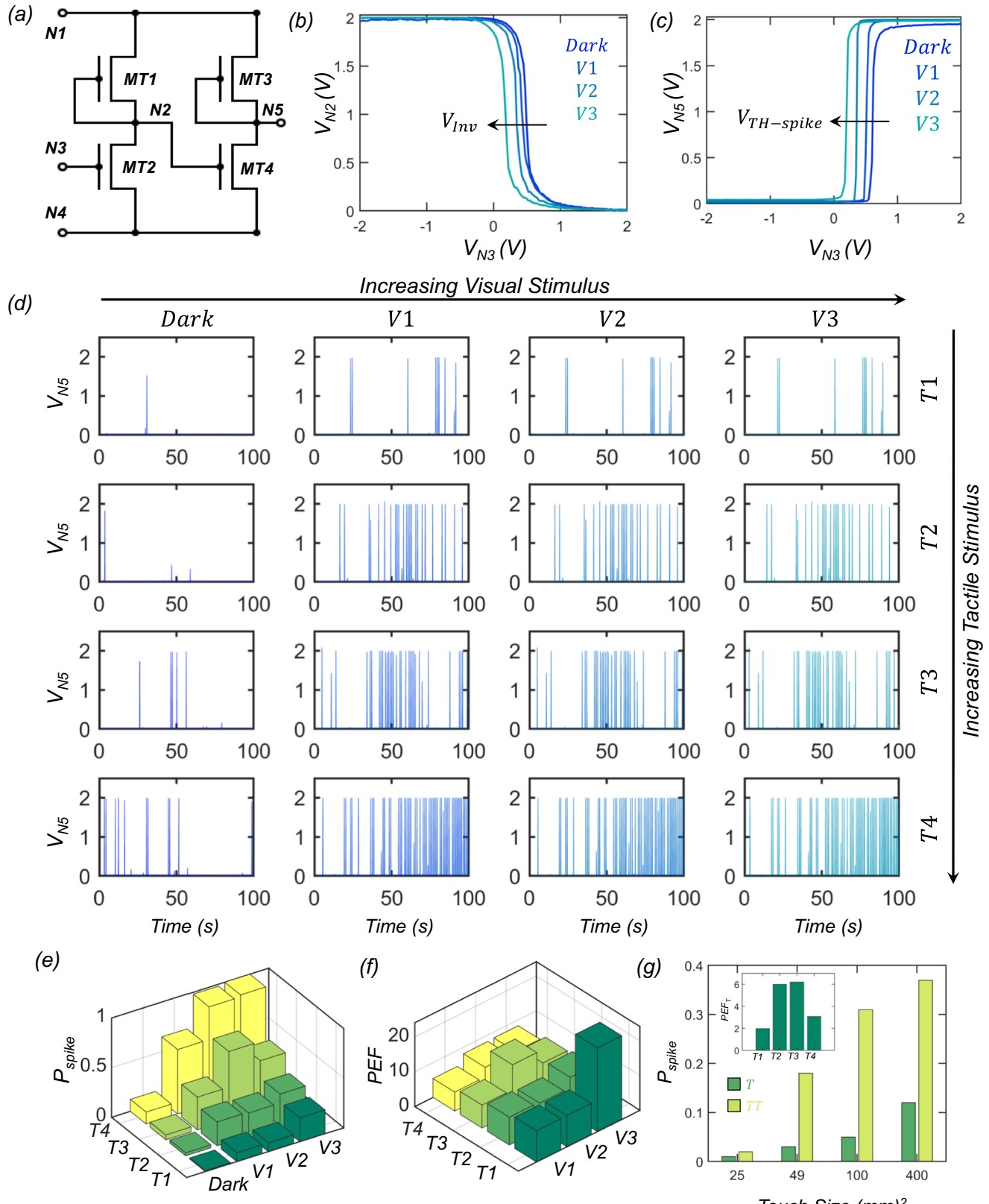

**Fig. 5 | Visuotactile spike encoding. a** Circuit diagram of the spike encoder comprising four monolayer MoS$_2$ photo-memtransistors (MT1–MT4). The circuit operates as a two-stage cascaded inverter. **b** Voltage, $V_{N2}$, measured at node N2, and **c** voltage, $V_{N5}$, measured at node N5 as a function of the input voltage, $V_{N3}$, applied to node N3 under dark condition and after exposure to different visual stimuli ($V$). The inversion threshold ($V_{Inv}$) of the first stage inverter and the spiking threshold ($V_{TH-spike}$) of the second stage inverter are functions of the applied visual stimulus. **d** Spiking output from the encoding circuit, recorded at node N5 in response to different tactile ($T$) and visual ($V$) stimuli. **e** Bar plot of the spiking probability ($P_{spike}$) for all combinations of $T$ and $V$. **f** Corresponding bar plots of the probability enhancement factor (PEF) obtained from the ratio of post-illumination $P_{spike}$ to pre-illumination $P_{spike}$. **g** Comparison of $P_{spike}$ obtained for single ($T$) and dual ($TT$) touches in the dark for all touch sizes. Inset shows the probability enhancement factor for unisensory tactile integration (PEF$_T$).

performance of multisensory neurons is discussed in Supplementary Fig. 16.

## Spike encoding of visuotactile cross-modal cues

The above demonstrations establish the fact that our bio-inspired visuotactile neuron exhibits all characteristic features of multisensory integration. However, unlike the brain, where information is encoded into digital spike trains, the response from our MN is analog. To convert the analog current responses into digital spikes, we use a circuit comprising four monolayer $MoS_2$ photo-memtransistors, MT1, MT2, MT3, and MT4, as shown in Fig. 5a (optical image shown in Fig. 1b). Since the gate and drain terminals are shorted for both MT1 and MT3, these photo-memtransistors are always in saturation and act as depletion loads. Figure 5b shows the voltage, $V_{N2}$, measured at node N2 as a function of the input voltage, $V_{N3}$, applied to node N3, i.e., the gate terminal of MT2, under dark condition and after exposure to different visual stimuli ($V$). For $V_{N3} = -2\,V$, MT2 is in the OFF-state (open circuit), pulling up $V_{N2}$ to $V_{DD} = 2\,V$, which is applied to the source terminal of MT1, i.e., node N1. Similarly, for $V_{N3} = 2\,V$, MT2 is in the ON-state (short circuit), pulling $V_{N2}$ down to $V_{GND} = 0\,V$, which is applied to the drain terminal of MT2, i.e., node N4. This explains why $V_{N2}$ switches from 2 V to 0 V as $V_{N3}$ is swept from $-2\,V$ to 2 V. In other words, MT1 and MT2 operate as a depletion mode inverter. In the transfer curve shown in Fig. 5b, the value of $V_{N3}$ at which $V_{N2} = V_{DD}/2$ is defined as the inversion threshold ($V_{Inv}$). Note that $V_{Inv}$ decreases monotonically with exposure to stronger visual stimuli. This is owing to the photogating effect, which results in a negative shift in $V_{TH}$ of MT2. Also note that MT3 and MT4 share a similar configuration and hence their role is to invert $V_{N2}$. Figure 5c shows the voltage, $V_{N5}$, measured at node N5 as a function of $V_{N3}$ under the same visual stimulus ($V$). Clearly, the circuit comprising MT1, MT2, MT3, and MT4 operates as a 2-stage cascaded inverter that can convert an analog input voltage, $V_{N3}$, into a digital output, $V_{N5}$; at the same time, this circuit offers visual memory, which in turn determines the spiking threshold, $V_{TH-spike}$, for the tactile stimulus (T).

Figure 5d shows the spiking response from the multisensory neural circuit for different combinations of $V$ and $T$. Clearly, the analog current response has now been converted into digital voltage spikes with the probability of spiking ($P_{spike}$) encoding the strengths of $V$ and $T$. Figure 5e shows the bar plot of extracted $P_{spike}$ for different combinations of $V$ and $T$. As expected, $P_{spike}$ shows a monotonic increase with increasing strength of $V$ and $T$ before saturating at the maximum value of 1. Figure 5f shows the bar plot of the probability enhancement factor (PEF), defined as the ratio of post-illumination $P_{spike}$ to pre-illumination $P_{spike}$, as a function of $V$ and $T$. Note that PEF » 1 for all combinations of $V$ and $T$, i.e., the spike encoder preserves the super-additive nature of multisensory integration. Also, PEF increases with increasing strength of $V$ as $V_{TH-spike}$ is reduced and can reach as high as ~24 for the weakest tactile cue, T1. Moreover, PEF is the largest for the smallest $T$ and decreases monotonically with increasing strength of $T$ for any given $V$. In other words, PEF also demonstrates the inverse effectiveness effect with $T$. Finally, Fig. 5g shows the bar plots for $P_{spike}$ for single ($T$) and dual ($TT$) tactile stimuli under dark conditions; the inset shows the probability enhancement factor for unisensory tactile integration ($PEF_T$) (see Supplementary Fig. 17 for spiking response from the multisensory neural circuit for dual touches of different strengths, $TT$). Note that $PEF_T$ » 1 for all $T$, confirming that the super-additive nature of unisensory integration is also preserved by the spike encoding circuit.

Also, note that the footprint of the visuotactile circuit is primarily determined by the dimensions of the 2D photo-memtransistor. Recently, we have shown ultra-scaled 2D devices with $L_{CH}$ down to 100 nm along with scaled contacts ($L_C$ down to 20 nm)[53]. There are several other reports in the literature that confirm the aggressive scalability of 2D devices[20,54,55]. The footprint of the visuotactile circuit

can therefore be made significantly smaller through device dimension scaling. However, the limiting factor is going to be the dimension of the photosensor, which will be determined by the diffraction limit of the light. For the visible spectrum, the dimensions of the photosensors have been stagnant at the micrometer scale or larger. Also note that instead of realizing the readout circuit using 2D photo-memtransistors, integration of 2D sensors with silicon CMOS is a viable alternative, but at the cost of increased processing and fabrication complexity while having no significant performance advantages. In fact, many recent reports highlight the energy and area benefits of using 2D memtransistors for in-sensor and near-sensor processing and storage since these multifunctional devices can be used as logic, memory, and sensing devices[25–27,34,35,37,51,52,56–62]. The motivation for our current work lies in the integration of these photo-memtransistors with tactile sensors to produce efficient and reliable multisensory integration. Finally, we envision that exploring arrays of multisensory neurons can enable more sophisticated visuotactile information processing.

## Discussion

In conclusion, we have realized a visuotactile MN comprising a triboelectric tactile sensor and a monolayer $MoS_2$ photo-memtransistor that offers all three characteristic features of multisensory integration, i.e., super-additive response, inverse effectiveness effect, and temporal congruency. We have also developed a visuotactile spike encoder circuit that converts the analog current response from the MN into digital spikes. We believe that our demonstration of multisensory integration will advance the field of neuromorphic and bio-inspired computing, which has primarily relied on unimodal sensory information processing to date. We also believe that the impact of multisensory integration can be far-reaching with applications in defense, space exploration, and many robotic and AI systems. Finally, the principles of multisensory integration can be expanded beyond visuotactile information processing to other sensory stimuli, including audio, olfactory, thermal, and gustatory stimuli.

## Methods

### Fabrication of local back-gate islands

To define the back-gate island regions, the substrate (285 nm $SiO_2$ on $p^{++}$-Si) was spin-coated with a bilayer resist stack consisting of EL6 and A3 resists at 4000 RPM for 45 s. After coating, the resists were baked at 150 °C for 90 s and 180 °C for 90 s, respectively. The bilayer resist was then patterned using e-beam lithography to define the islands and developed by immersing the substrate in a solution of 1:1 MIBK:IPA for 60 s, followed by immersion in 2-propanol (IPA) for 45 s. The back-gate electrode of 20/50 nm Ti/Pt was deposited using e-beam evaporation. The resist was removed using acetone and photoresist stripper (PRS 3000) and cleaned using IPA and de-ionized (DI) water. The atomic layer deposition (ALD) process was then implemented to grow 40 nm $Al_2O_3$, 3 nm $HfO_2$, and 7 nm $Al_2O_3$ on the entire substrate, including the island regions. To access the individual Pt back-gate electrodes, etch patterns were defined using the bilayer photoresist consisting of LOR 5A and SPR 3012. The bilayer photoresist was then exposed using a Heidelberg MLA 150 Direct Write exposure tool and developed using MF CD26 microposit. The 50 nm dielectric stack was subsequently dry-etched using a $BCl_3$ chemistry at 5 °C for 20 seconds, which was repeated four times to minimize heating in the substrate. Next, the photoresist was removed using the same process mentioned above to give access to the individual Pt electrodes.

### Large area monolayer $MoS_2$ film growth

Monolayer $MoS_2$ was grown on an epi-ready 2" c-sapphire substrate. The growth process utilized metal-organic chemical vapor deposition (MOCVD) in a cold-wall horizontal reactor equipped with an inductively heated graphite susceptor, as previously described[63]. Molybdenum hexacarbonyl ($Mo(CO)_6$) and hydrogen sulfide ($H_2S$) were utilized

as precursors for the process. Mo(CO)$_6$ was kept at 10 °C and 950 Torr in a stainless-steel bubbler, from which 0.036 sccm of the metal precursor was delivered for the growth. Simultaneously, 400 sccm of H$_2$S was introduced into the system. The deposition of MoS$_2$ took place at 1000 °C and 50 Torr in an H$_2$ atmosphere, allowing for the growth of monolayers within 18 minutes. Before initiating the growth, the substrate was initially heated to 1000 °C in an H$_2$ environment and held at this temperature for 10 minutes. After the growth process, the substrate was cooled down in an H$_2$S atmosphere to 300 °C to prevent the decomposition of the MoS$_2$ films. Further details can be found in our earlier work on this topic[20,36,64].

### MoS$_2$ film transfer to local back-gate islands

The fabrication process of the MoS$_2$ FETs involved transferring the MOCVD-grown monolayer MoS$_2$ film from the sapphire growth substrate to the SiO$_2$/p$^{++}$-Si substrate with local back-gate islands using a PMMA (polymethyl-methacrylate)-assisted wet transfer method. First, the MoS$_2$ film on the sapphire substrate was coated with PMMA through a spin-coating process, followed by baking at 180 °C for 90 s. Subsequently, the corners of the spin-coated film were carefully scratched using a razor blade, and the film was then immersed in a 1 M NaOH solution kept at 90 °C. Capillary action facilitated the entry of NaOH into the substrate/film interface, effectively separating the PMMA/MoS$_2$ film from the sapphire substrate. The separated film was thoroughly rinsed multiple times in a water bath to remove any remaining residues and contaminants. Finally, the film was transferred onto the SiO$_2$/p$^{++}$-Si substrate with local back-gate islands and baked at 50 °C and 70 °C for 10 min each, ensuring the removal of any moisture and PMMA was later removed using acetone, followed by cleaning with IPA.

### Fabrication of monolayer MoS$_2$ memtransistors

To define the channel regions for the MoS$_2$ memtransistors, the substrate underwent a series of steps. Firstly, it was spin-coated with PMMA and then baked at 180 °C for 90 s. The resist was then exposed to an electron beam (e-beam) and developed using 1:1 MIBK:IPA for 60 s, followed by immersion in IPA for 45 s. Subsequently, the monolayer MoS$_2$ film was etched using sulfur hexafluoride (SF$_6$) at 5 °C for 30 s. To remove the e-beam resist, the sample was rinsed in acetone and IPA. For defining the source and drain contacts, the sample was spin-coated with methyl methacrylate (MMA) followed by A3 PMMA. E-beam lithography was then used to pattern the source and drain contacts; development was performed using the same process detailed above. Next, e-beam evaporation was employed to deposit 40 nm of nickel (Ni) and 30 nm of gold (Au) as the contact metal. Finally, a lift-off process was performed to remove excess Ni/Au, leaving only the source/drain patterns. This lift-off was achieved by immersing the sample in acetone for 60 min, followed by IPA for another 10 min. Each island on the substrate contained one MoS$_2$ memtransistor, allowing individual gate control.

### Monolithic integration

The spike encoding circuit comprises 4 MoS$_2$ memtransistors. To establish the connections between the nodes, the substrate underwent several steps. Initially, it was spin-coated with MMA and PMMA. Next, e-beam lithography was employed, followed by the development of the same process detailed above. Subsequently, e-beam evaporation was performed to deposit 60 nm of Ni and 30 nm of Au. Finally, the e-beam resist was removed through a lift-off process achieved by immersing the substrate in acetone and IPA for 60 min and 10 min, respectively.

### Raman and photoluminescence (PL) spectroscopy

The MoS$_2$ film underwent Raman and PL spectroscopy using a Horiba LabRAM HR Evolution confocal Raman microscope equipped with a 532 nm laser. The laser power was adjusted to 34 mW and filtered down to 1.08 mW with a 3.2% filter. The Raman measurements were taken using an objective magnification of 100× and a numerical aperture of 0.9, while the grating spacing was set at 1800 gr/mm. For PL measurements, the grating spacing used was 300 gr/mm.

### Electrical characterization

The fabricated devices were electrically characterized using a Keysight B1500A parameter analyzer in a Lake Shore CRX-VF probe station, with measurements conducted under atmospheric conditions.

## Data availability

The datasets generated during and/or analyzed during the current study are available from the corresponding author upon reasonable request.

## Code availability

The codes used for plotting the data are available from the corresponding authors upon reasonable request.

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

## Acknowledgements

The work was supported by the Army Research Office (ARO) through Contract Number W911NF1810268 and National Science Foundation (NSF) through CAREER Award under Grant Number ECCS-2042154. The authors also express their gratitude for the 2D material support provided by Dr. Joan M. Redwing and the Pennsylvania State University 2D Crystal Consortium–Materials Innovation Platform (2DCCMIP).

## Author contributions

S.D. conceived the idea and designed the experiments. S.D., M.S., N.S., A.P., and H.R. performed the measurements, analyzed the data, discussed the results, and agreed on their implications. All authors contributed to the preparation of the paper.

## Competing interests

The authors declare no competing interests.
