## [Peer Review File · Nature Communications]

REVIEWER COMMENTS

Reviewer #1 (Remarks to the Author):

Multimodal sensory integration enhances the ability of detecting and recognizing objects and events. The authors report multimodal sensory integration using MoS₂ photoresponsive memtransistor and triboelectric tactile sensor and demonstrated spike encoding circuits. In addition, the authors emphasized the true characteristic features of multisensory integration, i.e. “super-additivity”, “inverse effectiveness effect” and “temporal congruency. This manuscript is interesting to the readers. There are several minor issues that need to be resolved before this manuscript is accepted, as follows.

(1) The reviewer suggests that the authors present a list table for comparing multimodal sensory design works using neuromorphic devices.

(2) More explanation on the mechanism of the photogating effect is needed.

(3) In fig. 2a & 2b, during the operation of the triboelectric tactile sensor, tactile response curve results of IDS produced some data below baselines, please explain here, what effect does this have on subsequent circuits? Can it be eliminated? Please provide detailed instructions for the operation of the triboelectric stimulation sensor.

(4) In the introduction section, the reviewer thinks that the authors need to add a bit of evidence for neuroscience background to demonstrate the importance of multimodal afferent integration. Some recent work on artificial neural devices with multisensory input was not mentioned. For example, Nature Electronics, 2020, 3(9): 563-570; Nature Communications 14, 1344 (2023); Research 2022, 2022, 9851843; Adv. Funct. Mater. 2023, 33, 2210119. “Here, we introduce a neuromorphic visuotactile device by integrating a triboelectric tactile sensor with a photosensitive monolayer MoS₂ memtransistor that can mimic the characteristic features and functionalities of a multisensory neuron.” The reviewer thinks that this sentence was not followed by a description of the full design of this work. For example, the spike coding circuit needs to be mentioned.

Reviewer #2 (Remarks to the Author):

Overall, the proposed visuotactile multisensory neuron including a triboelectric tactile sensor and monolayer MoS₂ photo memtransistor, has some novelty merits. Among the novel results are the

experimental demonstration of some key characteristics of such bioinspired multisensory integration, like the “super-additive” response, “inverse effectiveness effect” and “temporal congruency” based on the 2D Mo₂ and the triboelectric tactile sensor. There are a number of aspects that deserve clarification and improvement.

1) The triboelectric tactile sensor embodiment seems a basic laboratory setup that has little merits in terms of any future co-integration or even as performance and scaling of such visuotactile neuron. It is not clear why the authors selected such solution, which seems a bit far from the required merits of a technology to be published in a Nature journal. A co-integrated 2D material based tactile sensing principle would have been of much higher interest to show a more compact and scalable solution. Also, it is not clear how the ranges of generated gate voltages by the tactile pressure are designed to fit the particular sensor configuration of Fig. 1b) in terms of current modulation versus the optical modulation. For the publication, I would expect the authors to come with a co-integrated all-2D material based technology for both sensors and with an electrical design that would offer clear proof of balancing the importance of the generated electrical spiking signals.

2) The electrical MoS₂ transistor is well modeled and characterized, however it is not novel in itself. On the other hand, the authors mention that the encoding circuit was also built using MoS₂ memtransistors to convert analog IDS spikes to digital V_{out}. I would suggest to show a full top picture of the circuit and demonstrate that the spiking circuit and the sensing element are fully co-integrated. More discussion about the characterization, footprint and performance of such 2D sensor and 2D readout co-integration would be useful. Why the readout should be on 2D transistors and not simply made on CMOS with such 2D optical sensors integrated on top? The authors should provide more motivation and explanations about their system architecture choices and a perspective of their work, when moving from simple sensors into arrays of such memristive visuotactile neurons.

3) One would expect that the reported inverse effectiveness effect and super-additive response, as magnitude (aspects that I consider of interest and novel as experimental demonstration), would strongly depend on how the device is designed from the point of view of cross-sensitivity to light and to the electrical gating by the triboelectric tactile sensor. Such co-design of the gate stack of the transistor, of the sensitivity to light versus sensitivity to touch, that would correspond to a particular biological behavior is not discussed in-depth and it appears that the system is working, could be modeled but it is not clear how it should be designed and optimized in practice. I find necessary to have additional explanations about the cross-sensitivity co-design, beyond a simple electrical model of the transistor.

Reviewer #3 (Remarks to the Author):

The authors claim that all characteristic features of multisensory integration are demonstrated in their visuotactile multisensory neuron. The neuron comprises of a MoS₂ photo memtransistor and a triboelectric tactile sensor. The authors conducted systematic work on the neuron and explained the responses using empirical models. The results are very interesting, and the work may indeed inspire new approaches in the field of neuromorphic computing. Unfortunately, I cannot recommend acceptance of the manuscript for publication:

- The novelty of the manuscript is not argued sufficiently. Compared with the manuscript, a paper published on “Nano Research” (10.1007/s12274-023-5467-7) presents the same research idea, similar design/implementation, and comprehensive results/analysis. It is necessary to cite the published paper and highlight/justify the significance of this work.
- Page 4: Define MN (note this is defined in the caption of Figure 1).
- The morphology and structure of the devices are not adequately described. Is the MoS₂ single crystal? What are the dimensions of the devices? It may help to illustrate the structure of the device if the authors can present appropriate micrographs (both plan and cross-section views). The schematics in Figure 1 are useful, while it is not clear how the integration is implemented – this is not sufficiently described in the “Methods” section.
- It is not clear how the tactile stimuli were applied. The application of the “dual” touch needs clarifying. The authors need to discuss the reliability/reproducibility of these stimuli. What does it mean by “two identical touch inputs simultaneously”? How was this implemented?
- Page 6-7: “... to the surface charge density, which is strongly dependent on the surface contact area.” Please elaborate the dependence, i.e. how the charge density is modulated by the area.
- The results discussed in the manuscript seem from one neuron. The authors are encouraged to explore an ensemble of neurons. What are the crucial parameters of component devices (i.e. memtransistors and tactile sensors) to enable the reported behaviours of the multisensory integration, i.e. super-additive, inverse effectiveness and temporal congruency? What is the performance variability when material growth and device fabrication/evaluation are conducted with a set of optimised/controlled parameters?

Response to Reviewers

Reviewer 1:

Multimodal sensory integration enhances the ability of detecting and recognizing objects and events. The authors report multimodal sensory integration using MoS₂ photoresponsive memtransistor and triboelectric tactile sensor and demonstrated spike encoding circuits. In addition, the authors emphasized the true characteristic features of multisensory integration, i.e., “super-additivity”, “inverse effectiveness effect” and “temporal congruency. This manuscript is interesting to the readers. There are several minor issues that need to be resolved before this manuscript is accepted, as follows.

We are glad to know that the reviewer finds this work to be interesting and well-executed for demonstrating the true characteristic features of multisensory integration. We also thank the reviewer for their recommendation for publication of the manuscript in *Nature Communications*.

(1) The reviewer suggests that the authors present a list table for comparing multimodal sensory design works using neuromorphic devices.

We would like to thank the reviewer for their suggestion. The following discussion and a benchmarking table has been added to the revised manuscript and Supplementary Information.

While multisensory integration has been widely studied in cognitive, behavioral, and neurosciences, its benefits are not fully utilized in the fields of robotics, artificial intelligence, and neuromorphic computing. Note that there are some recent demonstrations of neuromorphic devices that can respond to more than one external stimuli. For example, Liu *et al.* [1] demonstrated a stretchable and photoresponsive nanowire transistor that can perceive both tactile and visual information, You *et al.* [2] demonstrated visuotactile integration using piezoresistor and MoS₂ field effect transistor (FET), Jiang *et al.* [3] used commercial sensors and spike encoding circuits to encode bimodal motion signals such as acceleration and angular speed and subsequently integrated the two using a dual gated MoS₂ FET, Wang *et al.* [4] demonstrated gesture recognition by integrating visual data with somatosensory data from stretchable sensors, Yu *et al.* [5] realized mechano-optic artificial synapse based on graphene/MoS₂ heterostructure and an integrated

triboelectric nanogenerator, and Sun *et al.* [6] reported an artificial reflex arc that senses visual and tactile information and processes using a self-powered optoelectronic perovskite (PSK) artificial synapse, and controls artificial muscular actions in response to environmental stimuli. Similarly, Chen *et al.* [7] reported a CsPbBr₃/TiO₂ based floating gate transistor, which can respond to both light and temperature, and Han *et al.* [8] proposed a fingerprint recognition system based on single-transistor neuron (1T-neuron) that can integrate visual and thermal stimuli. Finally, Yuan *et al.* [9] demonstrated VO₂ based artificial neuron that can encode illuminance, temperature, pressure, and curvature signals into spikes, and Liu *et al.* [10] reported an artificial autonomic nervous system to emulate the joint action of sympathetic and parasympathetic nerves on organs to control the contraction and relaxation of artificial pupils and visually simulate normal and abnormal heart rates. However, none of the neuromorphic devices mentioned above embrace the true characteristic features of multisensory integration, i.e., “super-additivity”, “inverse effectiveness effect” and “temporal congruency”. Furthermore, none of the above studies demonstrated spike encoding of multisensory information. Here, we introduce a neuromorphic visuotactile device by integrating a triboelectric tactile sensor with a photosensitive monolayer MoS₂ memtransistor that can mimic the characteristic features and functionalities of a multisensory neuron. A benchmarking table highlighting the advances made in this work over previous demonstrations on multisensory integration is shown in **Supplementary Table 1**.

No.	Materials and Devices Used		Demonstration of Key Multisensory Integration Features			Spike Encoding	Ref.
	Sensory input 1	Sensory input 2	Super-additivity	Inverse Effectiveness Effect	Temporal Congruency		
1.	MoS ₂ FET	Piezoresistor	✓	×	×	×	[2]
	Visual	Tactile					
2.	MoS ₂ FET	Commercial motion sensor	×	✓	×	×	[3]
	Visual	Vestibular					
3.	SONOS FinFET with Si ₃ N ₄ charge trap layer		×	×	×	×	[8]

	Visual	Thermal					
4.	MoS ₂ FET	PTFE/Cu triboelectric nanogenerator	×	×	×	×	[11]
	Visual	Tactile					
5.	CsPbBr ₃ /TiO ₂ floating gate FET		×	×	×	×	[12]
	Visual	Thermal					
6.	Self-powered perovskite (PSK) on ITO with SnO ₂ , Spiro-OMeTAD, and Au layers		×	×	×	×	[13]
	Visual	Tactile					
7.	P3HT/PEO NW FET, PVDF-HFP, EMIM-TFSI ion gel		×	×	×	×	[1]
	Visual	Thermal					
8.	Camera	SWCNT strain sensor	×	×	×	×	[4]
	Visual	Tactile					
9.	MOCVD MoS₂ FET	Kapton/Al based tactile sensor	✓	✓	✓	✓	This Work
	Visual	Tactile					

(2) More explanation on the mechanism of the photogating effect is needed.

We agree with the reviewer that more information on the mechanism of the photogating effect should be provided in the manuscript. We have added the following discussion in the **Supplementary Information 7** and referred to several previous literature to direct the reader to an in-depth understanding on the phenomenon of the photogating effect [14-17].

The photogating effect is a direct consequence of gate-tunable photocarrier trapping at the semiconductor/dielectric interface in field effect transistors (FETs) based on monolayer MoS₂, which is a direct bandgap semiconductor with $E_G = 1.84$ eV. When light illuminates the phototransistor, it generates photo carriers in the form of electron-hole pairs. If the MoS₂ FET is

biased in the on-state, photocarriers generated in the channel drift towards the respective electrodes under the applied source-to-drain bias resulting in non-persistent photoconductivity beyond the optical illumination. However, for illuminations in the off-state, noticeable shifts are observed in the threshold voltage (V_{TH}) of the device post-illumination, which is ascribed to the photo-gating effect i.e. trapping of photogenerated carrier at and near the MoS₂/Al₂O₃ interface. The photogating effect can also be explained using the energy band diagrams in **Fig. R1**. At equilibrium, i.e. in the absence of any gate bias, the trap states with energy levels above the Fermi energy (E_F) are empty, whereas the ones below E_F are filled. When the phototransistor is illuminated in the on-state or subthreshold region of the device operation, most trap states are below E_F making carrier trapping unlikely, which results in non-persistent photoconductivity beyond the optical illumination and the device returns to its initial state without any optical memory. However, when the phototransistor is illuminated in the off-state or depletion region of the device operation, most trap states are above E_F allowing carrier trapping at and near the MoS₂/Al₂O₃ interface. Negative shifts in V_{TH} indicate trapping of photo-generated holes. With longer illumination, more trap states are occupied leading to more shifts in V_{TH} . The detrapping process can be rather slow leading to persistent photocurrent MoS₂ phototransistor. More information on the phenomenon of the photogating effect can be found in our earlier work [14-17].

Figure R1. Energy band diagrams for explaining the photogating effect

(3) In fig. 2a & 2b, during the operation of the triboelectric tactile sensor, tactile response curve results of IDS produced some data below baselines, please explain here, what effect does this have on subsequent circuits? Can it be eliminated? Please provide detailed instructions for the operation of the triboelectric stimulation sensor.

The reviewer's observation is correct that tactile response does produce some I_{DS} values that are below the baseline. This can be explained using the operating mechanism of the triboelectric tactile sensor that involves contact electrification and electrostatic induction [18-20]. When the triboelectric material, Kapton, and the electrode, aluminum, come into contact, charge gets transferred from aluminum to Kapton by contact electrification due to Kapton's affinity to absorb electrons. This charge separation leads to an open circuit triboelectric potential, which is applied

to the gate terminal of the MoS₂ FET resulting in positive I_{DS} spikes above the baseline. As the touch input is removed, Kapton reverts to its original position, and an open circuit voltage of opposite polarity builds up, likely due to electrostatic induction, which leads to a decrease in I_{DS} below the baseline value. The typical voltage response profile from a triboelectric generator is shown in **Fig. R2**.

Note that, in order to mitigate the impact of

Figure R2. The typical voltage response profile from a triboelectric generator in response to touch stimulus.

I_{DS} values that are below the baseline the visuotactile encoding circuit was designed to convert the analog current responses into digital voltage spikes. As shown in **Fig. 5a** in the manuscript, the circuit comprised of four monolayer MoS₂ memtransistors, MT1, MT2, MT3, and MT4 (optical image can be found in **Fig. 1b**). Since the gate and drain terminals are shorted for both MT1 and MT3, these memtransistors are always in saturation and act as depletion loads. **Fig. 5b** shows the voltage, V_{N2} , measured at node, N2, as a function of the input voltage, V_{N3} , applied to node N3, i.e., the gate terminal of MT2, under dark condition. For $V_{N3} = -2$ V, MT2 is in the off-state (open circuit), pulling up V_{N2} to $V_{DD} = 2$ V, which is applied to the source terminal of MT1, i.e., at node N1. Similarly, for $V_{N3} = 2$ V, MT2 is in the ON-state (short circuit), pulling down V_{N2} to $V_{GND} = 0$ V, which is applied to the drain terminal of MT2, i.e., at node N4. This explains why V_{N2} switches from 2 V to 0 V as V_{N3} is swept from -2 V to 2 V. In other words, MT1 and MT2 operate as a depletion mode inverter. In the transfer curve in **Fig. 5b**, the value of V_{N3} at which $V_{N2} = V_{DD}/2$ is defined as the inversion threshold (V_{INV}). Also note that, MT3 and MT4 share a similar configuration and hence their role is to invert V_{N2} . **Fig. 5c** shows the voltage, V_{N5} , measured at

node, N_5 , as a function of V_{N_3} . Clearly, the circuit comprising of MT1, MT2, MT3, and MT4 operate as 2-stage cascaded inverter, which can convert analog input voltage, V_{N_3} , into digital output, V_{N_5} . **Fig. 5d** shows the spiking response from the multisensory neural circuit for different tactile stimuli, T in dark and under various optical stimulation, V . Clearly, the analog current response has now been converted into digital voltage spikes with the probability of spiking (P_{spike}) encoding the strengths of T . **Fig. R5e** shows the bar plot for extracted P_{spike} for different T . As expected, P_{spike} shows monotonic increase with increasing strengths of T before saturating at the maximum value of 1. This also explains the operation of the triboelectric stimulation sensor.

(4) In the introduction section, the reviewer thinks that the authors need to add a bit of evidence for neuroscience background to demonstrate the importance of multimodal afferent integration. Some recent work on artificial neural devices with multisensory input was not mentioned. For example, Nature Electronics, 2020, 3(9): 563-570; Nature Communications 14, 1344 (2023); Research 2022, 2022, 9851843; Adv. Funct. Mater.2023, 33, 2210119. "Here, we introduce a neuromorphic visuotactile device by integrating a triboelectric tactile sensor with a photosensitive monolayer MoS2 memtransistor that can mimic the characteristic features and functionalities of a multisensory neuron." The reviewer thinks that this sentence was not followed by a description of the full design of this work. For example, the spike coding circuit needs to be mentioned.

We would like to thank the reviewer for their comment. We have added the follow discussion in the revised introduction to include more evidence from neuroscience to demonstrate the importance of multimodal afferent integration.

Examples of multisensory information processing is abundant in nature. Dolphins, for instance, combine auditory cues derived from echoes with their visual system, enabling them to develop a comprehensive understanding of objects, distances, and shapes present in their environment. Honeybees communicate the whereabouts of food sources to their hive mates through intricate dances called "waggle dances." These dances incorporate visual cues, such as the angle and duration of the waggle, along with odor cues obtained from the nectar, effectively conveying information about the food source's distance and direction. Electric fish integrate sensory inputs from their electric sense, vision, and mechano-sensation to form a comprehensive perception of

their surroundings. While multisensory integration has been widely studied in cognitive, behavioral, and neurosciences, its benefits are not fully utilized in the fields of robotics, artificial intelligence, and neuromorphic computing.

As discussed in response to your earlier comment, we have cited the suggested papers and included in the introduction along with a benchmarking table highlighting the advances made in this work over previous demonstrations on multisensory integration. We have also mentioned the spike encoding circuit in the introduction to accurately reflect the full design of our work.

Reviewer #2 (Remarks to the Author):

Overall, the proposed visuotactile multisensory neuron including a triboelectric tactile sensor and monolayer MoS₂ photo memtransistor, has some novelty merits. Among the novel results are the experimental demonstration of some key characteristics of such bioinspired multisensory integration, like the “super-additive” response, “inverse effectiveness effect” and “temporal congruency” based on the 2D MoS₂ and the triboelectric tactile sensor. There are a number of aspects that deserve clarification and improvement.

We would like to thank the reviewer for appreciating the novelty of our work in the experimental demonstration of the three essential characteristic of multisensory integration.

1) The triboelectric tactile sensor embodiments seems a basic laboratory setup that has little merits in terms of any future co-integration or even as performance and scaling of such visuotactile neuron. It is not clear why the authors selected such a solution, which seems a bit far from the required merits of a technology to be published in a Nature journal. A co-integrated 2D material based tactile sensing principle would have been of much higher interest to show a more compact and scalable solution. Also, is not clear how the ranges of generated gate voltages by the tactile pressure are designed to fit the particular sensor configuration of Fig. 1b) in terms of current modulation versus the optical modulation. For the publication, I would expect the authors to come with a co-integrated all-2D material based technology for both sensors and with an electrical design that would offer clear proof of balancing the importance of the generated electrical spiking signals.

While we appreciate the reviewer’s opinion on co-integration of tactile and visual sensors, we feel it is not necessary for most practical applications. For example, if we want to equip a robot with multisensory capabilities, it is not essential for the tactile sensor to be located at the same position as the vision sensor. Biological entities, for example, humans observe visual stimuli using their eyes while simultaneously collecting tactile information primarily with their hands and other body parts. In fact, colocation of these sensory organs may come as a disadvantage when considering that optical evaluation is not possible without a finite separation distance from the subject while tactile interrogation requires direct physical contact with the subject. Therefore, we believe that

the basic laboratory setup has significant merits in highlighting the importance of multisensory integration and demonstrating the three essential features, namely “super additivity”, “inverse effectiveness effect”, and “temporal congruency”. To the best of our knowledge, and as confirmed by all three reviewers, the work presented in the manuscript is one of the most thorough experimental demonstration of multisensory integration along with numerical modeling, which justifies the publication of this work in Nature Communications.

Nevertheless, the reviewer has raised an important concern about the design of the sensor configuration to accommodate the gate voltage generated by the tactile sensor in terms of current modulation with respect to the optical modulation. We have added the following discussion in the **Supplementary Fig. 14 & 15**.

The overall design space is shown schematically in **Fig. R3**. Note that the visual response from monolayer MoS₂ phototransistor is obtained due to the photogating effect, which leads to a persistent shift in the threshold voltage (V_{TH-V}) and hence persistent photocurrent (I_{DS-V}) following **Eq. R1**. The external variables that influence V_{TH-V} and hence I_{DS-V} are 1) the strength of the optical illumination (I_{LED}), 2) duration of the optical illumination (t_{LED}), and 3) wavelength of the optical illumination (λ_{LED}). The biasing condition of the phototransistor can also influence the photo modulation. A comprehensive study on the impact of the above-mentioned factors

Figure R3. Design space for multisensory neuron.

associated with the optical stimuli has been reported in our earlier studies [14-17]. Similarly, the tactile response generates electrical impulse (V_{spike}) to the gate of the MoS₂ phototransistor leading to current spikes (I_{DS-T}) at the output following **Eq. R2**. The external variables that influence V_{spike} and hence I_{DS-T} are 1) surface charge density, which is strongly dependent on the surface contact area (T), and 2) the triboelectric material used (in our case Kapton). As described in the manuscript, V_{spike} follow a random Gaussian distribution with μ_T and σ_T as the mean and standard deviation, respectively. While the strength of the tactile stimulus (T) is captured through μ_T , the uncertainty associated with any triboelectric response is captured through σ_T . The dependence of μ_T on T is described using an empirical relationship. The fitting parameters, i.e., μ_{01} , T_{01} , μ_{02} , and T_{02} , are expected to show strong dependence on the gate capacitance (C_G) since V_{spike} is related to $Q_{tactile}/C_G$, where, $Q_{tactile}$ is the surface charge density that depends on T . Note that C_G depends on the thickness and dielectric constant of the gate insulator. Finally, both visual and tactile responses are influenced by the device dimensions such as channel length (L_{CH}), channel width (W_{CH}), and field effect carrier mobility (μ_{FE}). Therefore, in order to strike a balance between the visual and tactile current response, it is important to design the MoS₂ phototransistor-based multisensory neuron (MN) in such a way that V_{TH-V} and V_{spike} are of similar magnitudes. To do so, first it is important to have a knowledge of the application environment, i.e., the expected strength of the optical (I_{LED} , t_{LED} , λ_{LED}) and tactile (T) stimuli. Next, using the empirical and physics-based models described below and, in the manuscript, visual and tactile responses can be self-consistently and iteratively solved to arrive at the required device design dimensions.

$$I_{DS-V} = \frac{V_{DS}}{R_{CH}}; R_{CH} = \frac{L_{CH}}{W_{CH}\mu_N Q_{CH}}; Q_{CH} = C_G m \frac{k_B T}{q} \log \left[1 + \exp \left(\frac{V_{TH-V} - V_{TH}}{m k_B T_a / q} \right) \right] \quad [R1a]$$

$$V_{TH-V} = f(I_{LED}, t_{LED}, \lambda_{LED}) \quad [R1b]$$

$$I_{DS} = \frac{V_{DS}}{R_{CH}}; R_{CH} = \frac{L_{CH}}{W_{CH}\mu_N Q_{CH}}; Q_{CH} = C_G m \frac{k_B T}{q} \log \left[1 + \exp \left(\frac{V_{spike} - V_{TH}}{m k_B T_a / q} \right) \right] \quad [R2a]$$

$$V_{spike} = \text{Gaussian}(\mu_T, \sigma_T); \mu_T = \mu_{01} \left[1 - \exp \left(-\frac{T}{T_{01}} \right) \right] + \mu_{02} \left[1 - \exp \left(-\frac{T}{T_{02}} \right) \right] \quad [R2b]$$

At the same time, it is also important to understand how various device related parameters influences multisensory integration. **Fig. R4a-c**, respectively, show the dependence of MIF on μ_{FE} , V_{TH} , and SS for the weakest tactile and visual stimulus. Clearly, μ_{FE} has the least influence

Figure R4. Influence of device parameters on multisensory integration. Dependence of MIF on **a)** SS, **b)** μ_{FE} , and **c)** V_{TH} for the weakest tactile and visual stimulus. Clearly, μ_{FE} has the least influence on MIF since it is related to on-state performance of the memtransistor, whereas visuotactile responses are generated in the off-state of the memtransistor. Therefore, as expected, both V_{TH} , and SS have significant impact on MIF. Note that the dependence of MIF on various device related parameters can become a critical design consideration when an ensemble of multisensory neurons are present. **d)** Transfer characteristics of 100 multisensory neuron. Neuron-to-neuron variation in **e)** SS, **f)** V_{TH} and **g)** μ_{FE} . **h)** The projected neuron-to-neuron variation in MIF based on the models discussed in the manuscript.

on MIF since it is related to on-state performance of the memtransistor, whereas, visuotactile responses are generated in the off-state of the memtransistor. Therefore, as expected, both V_{TH} , and SS have significant impact on MIF with more positive V_{TH} and lower magnitude of SS leading to improved MIF. Note that the dependence of MIF on various device related parameters can become a critical design consideration when an ensemble of multisensory neurons are present. **Fig. R4d** shows the transfer characteristics of 100 multisensory neuron and **Fig. R4e-g**, respectively,

show the corresponding neuron-to-neuron variation in μ_{FE} , V_{TH} , and SS . The mean values for μ_{FE} , V_{TH} , and SS were found to be $\sim 12.6 \text{ cm}^2\text{V}^{-1}\text{s}^{-1}$, 0.6 V , and 231 mV/decade , respectively, with corresponding standard deviation values of $3.7 \text{ cm}^2\text{V}^{-1}\text{s}^{-1}$, 0.17 V , and 38 mV/decade , respectively. **Fig. R4h** shows the projected neuron-to-neuron variation in MIF based on the models discussed earlier. Note that the inherent variation in the tactile response was already built into the model described in **Eq. R2**. The mean value for MIF was found to be ~ 20 with a standard deviation of ~ 5 . It is possible to minimize the neuron-to-neuron variation through improving the synthesis, transfer, and cleanliness of process associated with the device fabrication.

2) The electrical MoS2 transistor is well modeled and characterized, however it is not novel in itself. On the other hand, the authors mention that the encoding circuit was also built using MoS2 memtransistors to convert analog I_{DS} spikes to digital V_{out} . I would suggest to show a full top picture of the circuit and demonstrate that the spiking circuit and the sensing element are full cointegrated. More discussion about the characterization, footprint and performance of such 2D sensor and 2D readout cointegration would be useful. Why the readout-out should be on 2D transistors and not simply made on CMOS with such 2D optical sensors integrated on top? The authors should provide more motivation and explanations about their system architecture choices and a perspective of their work, when moving from simple sensors into arrays of such memristive visuotactile neurons.

We would like to thank the reviewer for their appreciation of our modeling and through characterization of the MoS₂ transistor. We also appreciate reviewer's concern regarding the full top-view image of the circuit. Note that the optical image of the encoding circuit (2-stage cascaded inverter) was provided in **Fig 1b** (also see **Fig. R5**), which has been used to convert analog I_{DS} spikes to digital V_{out} . The circuit diagram is also shown in **Fig 1b** (also see **Fig. R5**), where we show how the tactile sensor is connected to the input of the spiking circuit while the output is recorded at the V_{out} terminal of the circuit.

We also agree with the reviewer that more discussion on characterization, footprint, and performance of 2D sensors is warranted. We have added the following discussion to the revised manuscript.

Figure R5. A bio-inspired multisensory visuotactile neuron comprising of a triboelectric tactile sensor connected to the gate terminal of a monolayer MoS_2 photo memtransistor along with the associated spike encoding circuit. Electrical impulses generated by the tactile sensor are transcribed into source-to-drain output current spikes (I_{DS}) by the MoS_2 photo memtransistor. On the other hand, visual stimuli are encoded into threshold voltage shift by exploiting photogating effect in monolayer MoS_2 photo memtransistor. The encoding circuit is also built using MoS_2 memtransistors to convert analog I_{DS} spikes to digital V_{out} .

Note that the footprint of the visuotactile circuit is primarily determined by the dimensions of the 2D memtransistor. Recently, we have shown ultra-scaled 2D devices with L_{CH} down to 100 nm along with scaled contacts (L_C down to 20 nm) [21]. There are several other reports in the literature, which confirms aggressive scalability of 2D devices [22-24]. Therefore, the footprint of the visuotactile circuit can be made significantly smaller through device dimension scaling. However, the limiting factor is going to be the dimension of the photosensor, which will be determined by the diffraction limit of the light. For the visible spectrum, the dimensions of the photosensors have been stagnant at the micrometer-scale or larger.

We also agree with the reviewer that a discussion on utilizing 2D readout circuit over CMOS-based readout circuit is warranted. We have added the following discussion to the revised manuscript.

Also note that instead of realizing the readout circuit using 2D memtransistors, integration of 2D sensor with CMOS is a viable alternative but at the cost of increased processing and fabrication complexity while having no significant performance advantages. In fact many recent reports

highlights the energy and area benefits of using 2D memtransistors for in-sensor and near-sensor processing and storage, since these multifunctional devices can be used as logic, memory, and sensing devices [14, 16, 17, 25-37]. The motivation of our current work lies in the integration of these phototransistors with tactile sensors to produce efficient and reliable multisensory integration. Finally, we envision that exploring arrays of multisensory neurons can enable more sophisticated visuotactile information processing.

3) One would expect that the reported inverse effectiveness effect and super-additive response, as magnitude (aspects that I consider of interest and novel as experimental demonstration), would strongly depend on how the device is designed from the point of view of cross-sensitivity to light and to the electrical gating by the triboelectric tactile sensor. Such co-design of the gate stack of the transistor, of the sensitivity to light versus sensitivity to touch, that would correspond to a particular biological behavior is not discussed in-depth and it appears that the system is working, could be modeled but is not clear how it should be designed and optimized in practice. I find necessary to have additional explanations about the cross-sensitivity co-design, beyond a simple electrical model of the transistor.

The reviewer asks about important considerations regarding the design of the device, and we believe we have addressed these concerns in our response to the first comment made by the reviewer.

Reviewer #3 (Remarks to the Author):

The authors claim that all characteristic features of multisensory integration are demonstrated in their visuotactile multisensory neuron. The neuron comprises of a MoS₂ photo memtransistor and a triboelectric tactile sensor. The authors conducted systematic work on the neuron and explained the responses using empirical models. The results are very interesting, and the work may indeed inspire new approaches in the field of neuromorphic computing. Unfortunately, I cannot recommend acceptance of the manuscript for publication:

We would like to express our gratitude to the reviewer for finding our work interesting and inspirational for advancing the field of neuromorphic computing.

- The novelty of the manuscript is not argued sufficiently. Compared with the manuscript, a paper published on “Nano Research” (10.1007/s12274-023-5467-7) presents the same research idea, similar design/implementation, and comprehensive results/analysis. It is necessary to cite the published paper and highlight/justify the significance of this work.*

We agree with the reviewer that we have underplayed the novelty of this work. We would also like to thank the reviewer for bringing the work published in “Nano Research” (10.1007/s12274-023-5467-7) to our attention. It is indeed excellent work on a similar idea with comprehensive analysis and results. We have cited the work in our revised manuscript. However, note that the work presented in “Nano Research” did not demonstrate all three characteristic features of multisensory integration, i.e., super-additive, inverse effectiveness, and temporal congruency. We could only find the “super-additive” feature being demonstrated in the “Nano Research” article in a rather superficial manner. Furthermore, the “Nano Research” work lacks device-level physical explanation behind the observation of multisensory integration effects. Therefore, our work can be considered a significant advance towards comprehensive experimental demonstration and numerical modeling of the phenomenon of multisensory visuotactile integration, which would justify its publication in Nature Communications. To further prove the novelty of our work, we have benchmarked our demonstration against previous works on multisensory integration, including the “Nano Research” work as shown below.

• Page 4: Define MN (note this is defined in the caption of Figure 1).

We would like to thank the reviewer for pointing this out. MN has been defined in the introduction of the revised manuscript.

• The morphology and structure of the devices are not adequately described. Is the MoS₂ single crystal? What are the dimensions of the devices? It may help to illustrate the structure of the device if the authors can present appropriate micrographs (both plan and cross-section views). The schematics in Figure 1 are useful, while it is not clear how the integration is implemented – this is not sufficiently described in the “Methods” section.

We agree with the reviewer’s comments. We have included the requested details in the revised manuscript and in the **Supplementary Information**.

Note that we have used large area monolayer MoS₂, which was grown using metal organic chemical vapor deposition (MOCVD) technique on a 2-inch c-plane sapphire substrate at 1000 °C. Extensive discussion on the synthesis of coalesced monolayer MoS₂ film can be found in the *Methods* section. Scanning transmission electron microscopy (STEM) was used to investigate the structure of the MoS₂ film used in this study and verify its quality. The same transfer technique used for device fabrication was used to transfer the as-grown film from its sapphire growth substrate to a TEM grid. A high-angle annular dark field (HAADF)-STEM image taken at an 80

Figure R6. a) Structure of MoS₂ viewed down its c-axis with atomic resolution high-angle annular dark field (HAADF)-scanning transmission electron microscope (STEM) imaging at an 80 kV accelerating voltage. Inset shows the atomic model of 2H-MoS₂ overlaid on the STEM image. b) Selected area electron diffraction (SAED) of the monolayer MoS₂, which reveals a uniform single-crystalline structure.

kV accelerating voltage is presented in **Fig. R6a**, showing the atomic structure of the MoS₂ film viewed down its c-axis. It can be clearly seen that the film possesses a crystalline 2H-MoS₂ structure with little-to-no point defects. This is further supported by the selected area electron diffraction (SAED) results shown in **Fig. R6b**, which show a uniform single-crystalline structure. Since these results are already reported in our previous work, we have cited the reference [14] in the revised manuscript.

We are sorry for missing the details on the device dimensions. The following discussion was added to the revised manuscript.

All MoS₂ FETs used in this study have channel length, $L_{CH} = 1 \mu\text{m}$ and channel width, $W_{CH} = 5 \mu\text{m}$ as shown using the plan-view optical micrograph in **Fig. R7a**. Cross-sectional TEM image in **Fig. R7b** shows the MoS₂ channel and the floating gate stack comprising of Al₂O₃/HfO₂/Al₂O₃ on Pt/Ti back-gate. Additionally, energy-dispersive X-ray spectroscopy (EDS) was used to analyze the elemental distribution of the stack, with the results being shown in **Fig. R7c**. A clear line of Mo and S atoms indicates the presence of the MoS₂ film, while the distribution of Al and Hf shows a clear delineation between the Al₂O₃ tunneling/blocking layers and the HfO₂ charge-trapping layer. These results confirm the structural integrity of the MoS₂ FET. We have added these details in the **Supplementary Fig 2**.

Figure R7. a) Plan-view optical micrograph and b) cross-sectional TEM image of monolayer MoS₂ based photo memtransistor showing the MoS₂ channel and the floating gate stack comprising of Al₂O₃/HfO₂/Al₂O₃ on Pt/Ti back-gate. c) Energy-dispersive X-ray spectroscopy (EDS) showing the elemental distribution of the stack. A clear line of Mo and S atoms indicates the presence of the MoS₂ film, while the distribution of Al and Hf shows a clear delineation between the Al₂O₃ tunneling/blocking layers and the HfO₂ charge-trapping layer

We also agree with the reviewer that a clearer description of how the integration is implemented is warranted. The following discussion was added as *Supplementary Fig 1*.

Fig. R8 shows the complete experimental set up. Note that our tactile sensor is comprised of a stack of commercially available Kapton and aluminum foil separated by an air gap, whereas PDMS stamps with different surface areas were prepared to serve as the touch stimuli. On the other hand, monolayer MoS₂ memtransistors serve as the optical sensor as well as used to construct the visuotactile spike encoding circuit. MoS₂ memtransistors, fabricated on a 1 cm × 1 cm substrate, are placed inside the Form Factor probe station, and are connected via probe tips/arms to external SMUs. Similarly, the top and bottom contacts of the tactile sensor are connected to SMUs via alligator clips. Finally, appropriate adapters are used to connect/short the relevant SMUs. A Keysight B1500 parameter analyzer is used for sourcing and measuring current/voltage through the respective SMUs.

Figure R8. Experimental set up for the demonstration of visuotactile multisensory integration.

• It is not clear how the tactile stimuli were applied. The application of the “dual” touch needs clarifying. The authors need to discuss the reliability/reproducibility of these stimuli. What does it mean by “two identical touch inputs simultaneously”? How was this implemented?

We apologize for the confusion. Optical images of various tactile inputs are shown in *Supplementary Fig 1*. Dual touch simply refers to using two identical touch inputs instead of one

and then conducting the experiments to see how the response changed when touch input area is doubled.

For better visualization of the tactile stimulation, we have included **Supplementary Video 1 & 2** for single and dual touches, respectively.

• Page 6-7: “... to the surface charge density, which is strongly dependent on the surface contact area.” Please elaborate the dependence, i.e. how the charge density is modulated by the area.

We would like to thank the reviewer for pointing this out. We are sorry for the mis-statement. The surface charge density is independent of the surface contact area (T), instead the total charge, $Q_{tactile}$, is proportional to T . Since V_{spike} is related to $Q_{tactile}/C_G$, where, C_G is the gate capacitance (C_G) of the MoS₂ FET, V_{spike} depends on T .

We have corrected the statement in the revised manuscript.

• The results discussed in the manuscript seem to be from one neuron. The authors are encouraged to explore an ensemble of neurons. What are the crucial parameters of component devices (i.e. memtransistors and tactile sensors) to enable the reported behaviors of the multisensory integration, i.e. super-additive, inverse effectiveness, and temporal congruency? What is the performance variability when material growth and device fabrication/evaluation are conducted with a set of optimized/controlled parameters?

The reviewer has made an excellent suggestion. We have now evaluated the impact of various device related parameters on multisensory integration for an ensemble of neurons. We have added the following discussion in the **Supplementary Fig. 14 & 15**.

The overall design space is shown schematically in **Fig. R9**. Note that the visual response from monolayer MoS₂ phototransistor is obtained due to the photogating effect, which leads to a persistent shift in the threshold voltage (V_{TH-V}) and hence persistent photocurrent (I_{DS-V}) following **Eq. R1**. The external variables that influence V_{TH-V} and hence I_{DS-V} are 1) the strength

of the optical illumination (I_{LED}), 2) duration of the optical illumination (t_{LED}), and 3) wavelength of the optical illumination (λ_{LED}). The biasing condition of the phototransistor can also influence the photo modulation. A comprehensive study on the impact of the above-mentioned factors associated with the optical stimuli has been reported in our earlier studies [14-17]. Similarly, the tactile response generates electrical impulse (V_{spike}) to the gate of the MoS₂ phototransistor leading

Figure R9. Design space for multisensory neuron.

to current spikes (I_{DS-T}) at the output following **Eq. R2**. The external variables that influence V_{spike} and hence I_{DS-T} are 1) surface charge density, which is strongly dependent on the surface contact area (T), and 2) the triboelectric material used (in our case Kapton). As described in the manuscript, V_{spike} follow a random Gaussian distribution with μ_T and σ_T as the mean and standard deviation, respectively. While the strength of the tactile stimulus (T) is captured through μ_T , the uncertainty associated with any triboelectric response is captured through σ_T . The dependence of μ_T on T is described using an empirical relationship. The fitting parameters, i.e., μ_{01} , T_{01} , μ_{02} , and T_{02} , are expected to show strong dependence on the gate capacitance (C_G) since V_{spike} is related to $Q_{tactile}/C_G$, where, $Q_{tactile}$ is the surface charge density that depends on T . Note that C_G depends on the thickness and dielectric constant of the gate insulator. Finally, both visual and tactile responses are influenced by the device dimensions such as channel length (L_{CH}), channel width (L_{CH}), and field effect carrier mobility (μ_{FE}). Therefore, in order to strike a balance between the visual and tactile current response, it is important to design the MoS₂ phototransistor-based

multisensory neuron (MN) in such a way that V_{TH-V} and V_{spike} are of similar magnitudes. To do so, first it is important to have a knowledge of the application environment, i.e., the expected strength of the optical ($I_{LED}, t_{LED}, \lambda_{LED}$) and tactile (T) stimuli. Next, using the empirical and physics-based models described below and, in the manuscript, visual and tactile responses can be self-consistently and iteratively solved to arrive at the required device design dimensions.

$$I_{DS-V} = \frac{V_{DS}}{R_{CH}}; R_{CH} = \frac{L_{CH}}{W_{CH}\mu_N Q_{CH}}; Q_{CH} = C_G m \frac{k_B T}{q} \log \left[1 + \exp \left(\frac{V_{TH-V} - V_{TH}}{m k_B T_a / q} \right) \right] \quad [R1a]$$

$$V_{TH-V} = f(I_{LED}, t_{LED}, \lambda_{LED}) \quad [R1b]$$

$$I_{DS} = \frac{V_{DS}}{R_{CH}}; R_{CH} = \frac{L_{CH}}{W_{CH}\mu_N Q_{CH}}; Q_{CH} = C_G m \frac{k_B T}{q} \log \left[1 + \exp \left(\frac{V_{spike} - V_{TH}}{m k_B T_a / q} \right) \right] \quad [R2a]$$

$$V_{spike} = Gaussian(\mu_T, \sigma_T); \mu_T = \mu_{01} \left[1 - \exp \left(-\frac{T}{T_{01}} \right) \right] + \mu_{02} \left[1 - \exp \left(-\frac{T}{T_{02}} \right) \right] \quad [R2b]$$

At the same time, it is also important to understand how various device related parameters influences multisensory integration. **Fig. R10a-c**, respectively, show the dependence of MIF on μ_{FE} , V_{TH} , and SS for the weakest tactile and visual stimulus. Clearly, μ_{FE} has the least influence on MIF since it is related to on-state performance of the memtransistor, whereas, visuotactile responses are generated in the off-state of the memtransistor. Therefore, as expected, both V_{TH} , and SS have significant impact on MIF with more positive V_{TH} and lower magnitude of SS leading to improved MIF . Note that the dependence of MIF on various device related parameters can become a critical design consideration when an ensemble of multisensory neurons are present. **Fig. R10d** shows the transfer characteristics of 100 multisensory neuron and **Fig. R10e-g**, respectively, show the corresponding neuron-to-neuron variation in μ_{FE} , V_{TH} , and SS . The mean values for μ_{FE} , V_{TH} , and SS were found to be $\sim 12.6 \text{ cm}^2\text{V}^{-1}\text{s}^{-1}$, 0.6 V , and 231 mV/decade , respectively, with corresponding standard deviation values of $3.7 \text{ cm}^2\text{V}^{-1}\text{s}^{-1}$, 0.17 V , and 38 mV/decade , respectively. **Fig. R10h** shows the projected neuron-to-neuron variation in MIF based on the models discussed earlier. Note that the inherent variation in the tactile response was already built into the model described in **Eq. 2**. The mean value for MIF was found to be ~ 20 with a standard deviation of ~ 5 . It is possible to minimize the neuron-to-neuron variation through improving the synthesis, transfer, and cleanliness of process associated with the device fabrication.

Figure R10. Influence of device parameters on multisensory integration. Dependence of MIF on **a)** SS, **b)** μ_{FE} , and **c)** V_{TH} for the weakest tactile and visual stimulus. Clearly, μ_{FE} has the least influence on MIF since it is related to on-state performance of the memtransistor, whereas visuotactile responses are generated in the off-state of the memtransistor. Therefore, as expected, both V_{TH} , and SS have significant impact on MIF. Note that the dependence of MIF on various device related parameters can become a critical design consideration when an ensemble of multisensory neurons are present. **d)** Transfer characteristics of 100 multisensory neuron. Neuron-to-neuron variation in **e)** SS, **f)** V_{TH} and **g)** μ_{FE} . **h)** The projected neuron-to-neuron variation in MIF based on the models discussed in the manuscript.

References

- [1] L. Liu, W. Xu, Y. Ni, Z. Xu, B. Cui, J. Liu, *et al.*, "Stretchable neuromorphic transistor that combines multisensing and information processing for epidermal gesture recognition," *ACS nano*, vol. 16, pp. 2282-2291, 2022.
- [2] J. You, L. Wang, Y. Zhang, D. Lin, B. Wang, Z. Han, *et al.*, "Simulating tactile and visual multisensory behaviour in humans based on an MoS₂ field effect transistor," *Nano Research*, 2023/03/06 2023.
- [3] C. Jiang, J. Liu, Y. Ni, S. Qu, L. Liu, Y. Li, *et al.*, "Mammalian-brain-inspired neuromorphic motion-cognition nerve achieves cross-modal perceptual enhancement," *Nature Communications*, vol. 14, p. 1344, 2023/03/11 2023.
- [4] M. Wang, Z. Yan, T. Wang, P. Cai, S. Gao, Y. Zeng, *et al.*, "Gesture recognition using a bioinspired learning architecture that integrates visual data with somatosensory data from stretchable sensors," *Nature Electronics*, vol. 3, pp. 563-570, 2020/09/01 2020.
- [5] J. Yu, X. Yang, G. Gao, Y. Xiong, Y. Wang, J. Han, *et al.*, "Bioinspired mechano-photonic artificial synapse based on graphene/MoS₂ heterostructure," *Science Advances*, vol. 7, p. eabd9117, 2021.
- [6] L. Sun, Y. Du, H. Yu, H. Wei, W. Xu, and W. Xu, "An artificial reflex arc that perceives afferent visual and tactile information and controls efferent muscular actions," *Research*, 2022.
- [7] G. Chen, X. Yu, C. Gao, Y. Dai, Y. Hao, R. Yu, *et al.*, "Temperature-controlled multisensory neuromorphic devices for artificial visual dynamic capture enhancement," *Nano Research*, 2022.
- [8] J.-K. Han, S.-Y. Yun, J.-M. Yu, S.-B. Jeon, and Y.-K. Choi, "Artificial Multisensory Neuron with a Single Transistor for Multimodal Perception through Hybrid Visual and Thermal Sensing," *ACS Applied Materials & Interfaces*, vol. 15, pp. 5449-5455, 2023/02/01 2023.
- [9] R. Yuan, Q. Duan, P. J. Tiw, G. Li, Z. Xiao, Z. Jing, *et al.*, "A calibratable sensory neuron based on epitaxial VO₂ for spike-based neuromorphic multisensory system," *Nature Communications*, vol. 13, p. 3973, 2022/07/08 2022.
- [10] L. Liu, Y. Ni, J. Liu, Y. Wang, C. Jiang, and W. Xu, "An Artificial Autonomic Nervous System That Implements Heart and Pupil as Controlled by Artificial Sympathetic and Parasympathetic Nerves," *Advanced Functional Materials*, vol. 33, p. 2210119, 2023.
- [11] J. Yu, X. Yang, G. Gao, Y. Xiong, Y. Wang, J. Han, *et al.*, "Bioinspired mechano-photonic artificial synapse based on graphene/MoS₂ heterostructure," *Science Advances*, vol. 7, p. eabd9117, 2021.
- [12] G. Chen, X. Yu, C. Gao, Y. Dai, Y. Hao, R. Yu, *et al.*, "Temperature-controlled multisensory neuromorphic devices for artificial visual dynamic capture enhancement," *Nano Research*, vol. 16, pp. 7661-7670, 2023/05/01 2023.
- [13] L. Sun, Y. Du, H. Yu, H. Wei, W. Xu, and W. Xu, "An Artificial Reflex Arc That Perceives Afferent Visual and Tactile Information and Controls Efferent Muscular Actions," *Research*, vol. 2022, 2022.
- [14] A. Dodda, D. Jayachandran, A. Pannone, N. Trainor, S. P. Stepanoff, M. A. Steves, *et al.*, "Active pixel sensor matrix based on monolayer MoS₂ phototransistor array," *Nature Materials*, vol. 21, pp. 1379-1387, 2022/12/01 2022.
- [15] S. Subbulakshmi Radhakrishnan, S. Chakrabarti, D. Sen, M. Das, T. F. Schranhamer, A. Sebastian, *et al.*, "A Sparse and Spike-Timing-Based Adaptive Photoencoder for

- Augmenting Machine Vision for Spiking Neural Networks," *Advanced Materials*, vol. 34, p. 2202535, 2022.
- [16] S. Subbulakshmi Radhakrishnan, A. Dodda, and S. Das, "An All-in-One Bioinspired Neural Network," *ACS Nano*, vol. 16, pp. 20100-20115, 2022/12/27 2022.
- [17] A. Dodda, D. Jayachandran, S. Subbulakshmi Radhakrishnan, A. Pannone, Y. Zhang, N. Trainor, *et al.*, "Bioinspired and Low-Power 2D Machine Vision with Adaptive Machine Learning and Forgetting," *ACS Nano*, vol. 16, pp. 20010-20020, 2022/12/27 2022.
- [18] G. Zhu, C. Pan, W. Guo, C.-Y. Chen, Y. Zhou, R. Yu, *et al.*, "Triboelectric-Generator-Driven Pulse Electrodeposition for Micropatterning," *Nano Letters*, vol. 12, pp. 4960-4965, 2012/09/12 2012.
- [19] S. Niu, S. Wang, L. Lin, Y. Liu, Y. S. Zhou, Y. Hu, *et al.*, "Theoretical study of contact-mode triboelectric nanogenerators as an effective power source," *Energy & Environmental Science*, vol. 6, pp. 3576-3583, 2013.
- [20] Z. L. Wang, L. Lin, J. Chen, S. Niu, and Y. Zi, "Triboelectric Nanogenerator: Vertical Contact-Separation Mode," in *Triboelectric Nanogenerators*, ed Cham: Springer International Publishing, 2016, pp. 23-47.
- [21] T. F. Schranghamer, N. U. Sakib, M. U. K. Sadaf, S. Subbulakshmi Radhakrishnan, R. Pendurthi, A. D. Agyapong, *et al.*, "Ultrascaled Contacts to Monolayer MoS₂ Field Effect Transistors," *Nano Letters*, vol. 23, pp. 3426-3434, 2023/04/26 2023.
- [22] Q. Smets, G. Arutchelvan, J. Jussot, D. Verreck, I. Asselberghs, A. N. Mehta, *et al.*, "Ultrascaled MOCVD MoS₂ MOSFETs with 42nm contact pitch and 250 μ A/ μ m drain current," in *2019 IEEE International Electron Devices Meeting (IEDM)*, 2019, pp. 23.2. 1-23.2. 4.
- [23] A. Sebastian, R. Pendurthi, T. H. Choudhury, J. M. Redwing, and S. Das, "Benchmarking monolayer MoS₂ and WS₂ field-effect transistors," *Nature Communications*, vol. 12, p. 693, 2021/01/29 2021.
- [24] S. Das, A. Sebastian, E. Pop, C. J. McClellan, A. D. Franklin, T. Grasser, *et al.*, "Transistors based on two-dimensional materials for future integrated circuits," *Nature Electronics*, vol. 4, pp. 786-799, 2021/11/01 2021.
- [25] A. Sebastian, A. Pannone, S. S. Radhakrishnan, and S. Das, "Gaussian synapses for probabilistic neural networks," *Nature communications*, vol. 10, pp. 1-11, 2019.
- [26] D. Jayachandran, A. Oberoi, A. Sebastian, T. H. Choudhury, B. Shankar, J. M. Redwing, *et al.*, "A low-power biomimetic collision detector based on an in-memory molybdenum disulfide photodetector," *Nature Electronics*, 2020.
- [27] A. J. Arnold, A. Razavieh, J. R. Nasr, D. S. Schulman, C. M. Eichfeld, and S. Das, "Mimicking Neurotransmitter Release in Chemical Synapses via Hysteresis Engineering in MoS₂ Transistors," *ACS nano*, vol. 11, pp. 3110-3118, 2017.
- [28] H. Ravichandran, Y. Zheng, T. F. Schranghamer, N. Trainor, J. M. Redwing, and S. Das, "A Monolithic Stochastic Computing Architecture for Energy Efficient Arithmetic," *Advanced Materials*, vol. 35, p. 2206168, 2023.
- [29] D. Jayachandran, A. Pannone, M. Das, T. F. Schranghamer, D. Sen, and S. Das, "Insect-Inspired, Spike-Based, in-Sensor, and Night-Time Collision Detector Based on Atomically Thin and Light-Sensitive Memtransistors," *ACS Nano*, 2022/12/30 2022.
- [30] A. Sebastian, R. Pendurthi, A. Kozhakhmetov, N. Trainor, J. A. Robinson, J. M. Redwing, *et al.*, "Two-dimensional materials-based probabilistic synapses and reconfigurable neurons for measuring inference uncertainty using Bayesian neural networks," *Nature communications*, vol. 13, pp. 1-10, 2022.

- [31] S. Chakrabarti, A. Wali, H. Ravichandran, S. Kundu, T. F. Schranghamer, K. Basu, *et al.*, "Logic Locking of Integrated Circuits Enabled by Nanoscale MoS₂-Based Memtransistors," *ACS Applied Nano Materials*, 2022/10/04 2022.
- [32] S. S. Radhakrishnan, S. Chakrabarti, D. Sen, M. Das, T. F. Schranghamer, A. Sebastian, *et al.*, "A Sparse and Spike-timing-based Adaptive Photo Encoder for Augmenting Machine Vision for Spiking Neural Networks," *Advanced Materials*, p. 2202535.
- [33] Y. Zheng, H. Ravichandran, T. F. Schranghamer, N. Trainor, J. M. Redwing, and S. Das, "Hardware implementation of Bayesian network based on two-dimensional memtransistors," *Nature Communications*, vol. 13, p. 5578, 2022/09/23 2022.[34] A. Dodda, N. Trainor, J. Redwing, and S. Das, "All-in-one, bio-inspired, and low-power crypto engines for near-sensor security based on two-dimensional memtransistors," *Nature communications*, vol. 13, pp. 1-12, 2022.
- [35] R. Pendurthi, D. Jayachandran, A. Kozhakhmetov, N. Trainor, J. A. Robinson, J. M. Redwing, *et al.*, "Heterogeneous Integration of Atomically Thin Semiconductors for Non-von Neumann CMOS," *Small*, p. 2202590, 2022.
- [36] A. Sebastian, S. Das, and S. Das, "An Annealing Accelerator for Ising Spin Systems Based on In-Memory Complementary 2D FETs," *Advanced Materials*, vol. 34, p. 2107076, 2022/01/01 2022.
- [37] A. Sebastian, A. Pannone, S. Subbulakshmi Radhakrishnan, and S. Das, "Gaussian synapses for probabilistic neural networks," *Nat Commun*, vol. 10, p. 4199, Sep 13 2019.

REVIEWER COMMENTS

Reviewer #1 (Remarks to the Author):

The author's revised manuscript has been greatly improved, and the issues raised have been addressed, but there is a little confusion to be solved. In the introduction part, "Furthermore, none of the above studies demonstrated spike encoding of multisensory information." The reviewer feels that this description is inaccurate. In fact, Information coding has been mentioned in the report of the artificial reflex arc of dual neural pathways (Sun et al. Research. 2022:2022, the visual and somatosensory information is encoded as impulse spikes). The reviewer suggests that the author revise this, as well as the corresponding Supplementary Table 1 of Demonstration of Multisensory Integration.

Reviewer #2 (Remarks to the Author):

Overall, the authors have satisfactorily addressed the questions and improved the paper. The paper holds merits in terms of novelty and even if the reported performance of the visuotactile neuron and triboelectric tactile sensor is not high, the "super-additive" response is clearly demonstrated.

The following minor clarifications/changes are recommended:

(i) The authors provided good answers to my question on the co-design to adapt the gate voltage generated by the tactile sensor in terms of current modulation with respect to the optical modulation - the values of mobility and subthreshold slope are not really at the top of state of the art and the neuron-to-neuron variation is still very large. The authors mention very generically that some improvements can be done by certain processing techniques. In this case the question to be commented remains: is the MoS₂ phototransistor (monolayer, bilayer, multilayer, not clear!) the preferable solutions or other technological options (as semiconducting material and down-scaling) would better serve the implementations. More detailed comments and targeted figures of merit motivating the MoS₂ choice at device level would be welcome.

(ii) One important question for such system relates to the drift and temperature stability as well as the temperature calibration needed for the multisensory system in practical applications. The authors are citing other works in which temperature is co-monitored and the effect considered. Some of the reported variability could be even due to temperature variations among multiple

experiments. Any existing data, discussion or insights about this aspects would add value to the practical applications of this work.

Reviewer #3 (Remarks to the Author):

The authors have addressed all of my comments. I am happy to recommend acceptance for publication.

Response to Reviewers

Reviewer 1:

The author's revised manuscript has been greatly improved, and the issues raised have been addressed, but there is a little confusion to be solved. In the introduction part, "Furthermore, none of the above studies demonstrated spike encoding of multisensory information." The reviewer feels that this description is inaccurate. In fact, Information coding has been mentioned in the report of the artificial reflex arc of dual neural pathways (Sun et al. Research. 2022:2022, the visual and somatosensory information is encoded as impulse spikes). The reviewer suggests that the author revise this, as well as the corresponding Supplementary Table 1 of Demonstration of Multisensory Integration.

We greatly appreciate the reviewer's high praise for the improvement of the manuscript. We would also like to thank the reviewer for acknowledging that their concerns have been addressed. Furthermore, we appreciate the reviewer for pointing out the information encoding presented in the referenced manuscript. We have revised the manuscript and Supplementary table accordingly.

Reviewer 2:

Overall, the authors have satisfactorily addressed the questions and improved the paper. The paper holds merits in terms of novelty and even if the reported performance of the visuotactile neuron and triboelectric tactile sensor is not high, the “super-additive” response is clearly demonstrated.

We would like to express our gratitude to the reviewer for stating that their questions have been addressed satisfactorily and for finding the current version of the manuscript an improvement compared to the previous version. We would like to further thank the reviewer for appreciating the novelty of our work.

The following minor clarifications/changes are recommended:

1) The authors provided good answers to my question on the co-design to adapt the gate voltage generated by the tactile sensor in terms of current modulation with respect to the optical modulation - the values of mobility and subthreshold slope are not really at the top of state of the art and the neuron-to-neuron variation is still very large. The authors mention very generically that some improvements can be done by certain processing techniques. In this case the question to be commented remains: is the MoS₂ phototransistor (monolayer, bilayer, multilayer, not clear!) the preferable solutions or other technological options (as semiconducting material and down-scaling) would better serve the implementations. More detailed comments and targeted figures of merit motivating the MoS₂ choice at device level would be welcome.

We are glad to know that we have been able to provide good answers to the reviewer’s questions on the co-design to adapt the gate voltage generated by the tactile sensor for current modulation with respect to the optical modulation. We address the remaining comments raised by the reviewer below. We have also included the discussion as ***Supplementary Information 3***.

We agree with the reviewer’s statement that our field effect carrier mobility and subthreshold slope values are not the best values reported in literature. However, we are at par with most reports. The champion mobility value for monolayer MoS₂ grown by chemical vapor deposition (CVD) was found to be $\sim 55 \text{ cm}^2\text{V}^{-1}\text{s}^{-1}$ at room temperature with bismuth (Bi) as the contact metal [1]. Another recent study, where antimony (Sb) was used as the contact metal, reported mobility value of ~ 50

$\text{cm}^2\text{V}^{-1}\text{s}^{-1}$ for CVD grown monolayer MoS_2 [2]. In contrast, the highest mobility value for metal organic CVD (MOCVD) grown MoS_2 used in this study was found to be $\sim 26 \text{ cm}^2\text{V}^{-1}\text{s}^{-1}$. In an earlier study, we reported champion mobility value of $46 \text{ cm}^2\text{V}^{-1}\text{s}^{-1}$, which is comparable to the current state-of-the-art values [3]. Note that the mobility of MOCVD grown monolayer MoS_2 with typical grain size of $\sim 1 \mu\text{m}$ is expected to be lower than CVD grown single crystal monolayer MoS_2 . However, MOCVD is considered a preferable synthesis technique for manufacturing purposes due to its ability to produce conformal films, as opposed to the formation of large triangular flakes typically obtained using CVD [4-6]. Nevertheless, since our MoS_2 -based multisensory neuron (MN) is mostly operated in the subthreshold regime, mobility plays a less significant role in multisensory integration factor (*MIF*) as shown in **Fig. R1a** (also see **Supplementary Fig. 15b**).

The reviewer also mentions the subthreshold slope (*SS*) not being at the state-of-the-art level. Subthreshold slope can be explained using the following equation.

$$SS = \frac{mk_B T}{q} \ln(10); m = \left(1 + \frac{C_S}{C_{OX}} + \frac{C_{IT}}{C_{OX}}\right); C_{IT} = qD_{IT} \quad [R1]$$

Here k_B is the Boltzmann constant, T is the temperature, q is the electron charge, m is the body factor, C_S is the semiconductor capacitance, C_{OX} is the oxide capacitance, C_{IT} is the interface trap capacitance, and D_{IT} is the interface trap density. For ultra-thin body (UTB) semiconductors such as monolayer MoS_2 , $C_S = 0$, and for a clean semiconductor interface, $C_{IT} \ll C_{OX}$ which ensures that $m = 1$ and $SS = 60 \text{ mVdec}^{-1}$. In this work, since the reported mean SS value is 255 mVdec^{-1} , C_{IT} is finite and using **Eq. R1**, we can extract the mean value for D_{IT} to be $\sim 4.5 \times 10^{12} \text{ eV}^{-1}\text{cm}^{-2}$. In comparison, D_{IT} value for the state-of-the-art UTB Si FETs is found to be $\sim 1.5 \times 10^{12} \text{ eV}^{-1}\text{cm}^{-2}$ leading to an SS of 80 mVdec^{-1} [7], for an effective oxide thickness (EOT) of 4 nm. Therefore, our reported mean D_{IT} value for the monolayer MoS_2 FET is comparable to the state-of-the-art Si FETs. Therefore, it is possible to achieve better SS by reducing the EOT, which in our case is 20 nm. As mentioned earlier, the MN operates in the off-state, therefore, there will be a significant effect of SS on the multisensory integration factor (*MIF*). In order to find the optimum SS values and other device related parameters, we can self-consistently and iteratively solve the empirical and physics-based models described below.

$$I_{DS-V} = \frac{V_{DS}}{R_{CH}}; R_{CH} = \frac{L_{CH}}{W_{CH}\mu_{MN}Q_{CH}}; Q_{CH} = C_G m \frac{k_B T}{q} \log \left[1 + \exp \left(\frac{V_{TH-V} - V_{TH}}{m k_B T a / q} \right) \right] \quad [R2a]$$

$$V_{TH-V} = f(I_{LED}, t_{LED}, \lambda_{LED}) \quad [R2b]$$

$$I_{DS} = \frac{V_{DS}}{R_{CH}}; R_{CH} = \frac{L_{CH}}{W_{CH}\mu_N Q_{CH}}; Q_{CH} = C_G m \frac{k_B T}{q} \log \left[1 + \exp \left(\frac{V_{spike} - V_{TH}}{m k_B T_a / q} \right) \right] \quad [R3a]$$

$$V_{spike} = Gaussian(\mu_T, \sigma_T); \mu_T = \mu_{01} \left[1 - \exp \left(-\frac{T}{T_{01}} \right) \right] + \mu_{02} \left[1 - \exp \left(-\frac{T}{T_{02}} \right) \right] \quad [R3b]$$

For the weakest visual and tactile stimulus, we can see the dependence of SS to multisensory integration factor (MIF) in **Fig. R1b** (also see **Supplementary Fig. 15a**). Here, we can observe that in order to achieve multisensory integration, SS values should stay below 400 mVdec^{-1} to see a significant sensory response enhancement and MIF starts rising exponentially as SS values climb further down.

Next, we would like to address the reviewer's concern on neuron-to-neuron variation. Here, we would like to point out that most of the neurons exhibited SS values within 350 mVdec^{-1} range showing that they are all capable of demonstrating multisensory integration. The reviewer also raises an important concern regarding the MoS_2 in this study being monolayer or not. Raman and photoluminescence (PL) spectra of the MoS_2 used in this study was obtained using a 532 nm laser as shown in **Fig. R1c,d** (also see **Supplementary Fig. 1a,b**). Raman spectra of the monolayer MoS_2 shows the in-plane E_{2g} and out-of-plane A_{1g} peaks at 384.24 and 401.53 cm^{-1} , respectively, with a peak separation of $\approx 17 \text{ cm}^{-1}$, which corresponds to having a monolayer MoS_2 film. Photoluminescence (PL) spectra for the MoS_2 shows a peak at 1.83 eV , which also confirms having a monolayer film.

Finally, the reviewer raises concern about the choice of MoS_2 as the semiconducting material compared to other technological options when down scaling is also considered that would better serve the multisensory neuron implementation. Current complementary metal-oxide-semiconductor technology has already reached sub 10 nm technology nodes with further scaling becoming increasingly challenging because of the bulk nature of Si which causes increased scattering of charge carriers leading to severe mobility degradation with further aggressive scaling [8, 9]. In contrast to this, 2D materials (MoS_2 , WS_2 , WSe_2 etc.) can provide the ultimate channel thickness which is under 1 nm . Along with that, 2D based transistors can be used for both logic and memory applications which is a major bottleneck in the current CMOS technology [10]. Along with that, the material of our choice, MoS_2 has already been used for chemical [11], gas [12],

acoustic [13], and temperature sensing [14] along with visual sensing [15] and, therefore, is the perfect candidate when multisensory integration is considered.

Figure R1. Dependence of MIF on **a)** μ_{FE} , and **b)** SS for the weakest tactile and visual stimulus. **c)** Raman spectra of the monolayer MoS₂, obtained using a 532 nm laser, shows the in-plane E_{2g} and out-of-plane A_{1g} peaks at 384.24 and 401.53 cm⁻¹, respectively, with a peak separation of ≈ 17 cm⁻¹, which corresponds to having a monolayer MoS₂ film. **d)** Photoluminescence (PL) spectra for monolayer MoS₂ with a peak at 1.83 eV, which also confirms having a monolayer film.

2) One important question for such system relates to the drift and temperature stability as well as the temperature calibration needed for the multisensory system in practical applications. The authors are citing other works in which temperature is co-monitored and the effect considered. Some of the reported variability could be even due to temperature variations among multiple experiments. Any existing data, discussion or insights about this aspects would add value to the practical applications of this work.

We would like to thank the reviewer for raising some important points regarding the drift, temperature stability, and temperature calibration needed for the multisensory system. Note that all our experiments were conducted in a controlled laboratory environment where temperature is maintained at 293 ± 2 K. Therefore, we can eliminate temperature variation as the primary reason behind neuron-to-neuron variation as well as variations among multiple experiments. However, we do agree with the reviewer that for real-life applications temperature can play an important role in the performance of the MN. We have now performed additional experiments to elucidate the impact of temperature on monolayer MoS₂ FETs. **Fig. R2a** shows the transfer characteristics of a representative monolayer MoS₂ FET measured at different temperatures (T) ranging from 30°C to 60°C in the step of 10°C. **Fig. R2b** shows the extracted SS as a function of T . SS value is found to change by only 27%. The temperature dependence of SS will directly influence the MIF . Therefore, depending on the targeted application it may be necessary to employ standard techniques to mitigate temperature related drift. This can involve using temperature sensors to measure the ambient temperature and adjusting the circuit parameters or calibration values accordingly. Alternatively, circuit design techniques such as negative feedback, voltage references, and compensation networks can be used to mitigate temperature-related drift. These techniques help stabilize the circuit's performance despite temperature variations. Yet another way is to incorporate guard rings and shielding techniques to minimize the impact of temperature. By using one of more of these techniques, it is possible to mitigate the impact temperature-related drift on MN. We have included these results and discussion in **Supplementary Fig. 16**.

Figure R2. a) Transfer characteristics of a representative monolayer MoS₂ FET measured at different temperatures (T) ranging from 30°C to 60°C in the step of 10°C. **b)** Subthreshold slope (SS) as a function of increasing temperature.

Reviewer 3:

The authors have addressed all of my comments. I am happy to recommend acceptance for publication.

We are glad to know that we have addressed all of the reviewer's comments and we would like to express our sincere gratitude to the reviewer for recommending acceptance for publication.

References

- [1] P.-C. Shen, C. Su, Y. Lin, A.-S. Chou, C.-C. Cheng, J.-H. Park, *et al.*, "Ultralow contact resistance between semimetal and monolayer semiconductors," *Nature*, vol. 593, pp. 211-217, 2021/05/01 2021.
- [2] W. Li, X. Gong, Z. Yu, L. Ma, W. Sun, S. Gao, *et al.*, "Approaching the quantum limit in two-dimensional semiconductor contacts," *Nature*, vol. 613, pp. 274-279, 2023/01/01 2023.
- [3] A. Sebastian, R. Pendurthi, T. H. Choudhury, J. M. Redwing, and S. Das, "Benchmarking monolayer MoS₂ and WS₂ field-effect transistors," *Nature Communications*, vol. 12, p. 693, 2021/01/29 2021.
- [4] L. Yu, D. El-Damak, U. Radhakrishna, X. Ling, A. Zubair, Y. Lin, *et al.*, "Design, Modeling, and Fabrication of Chemical Vapor Deposition Grown MoS₂ Circuits with E-Mode FETs for Large-Area Electronics," *Nano Letters*, vol. 16, pp. 6349-6356, 2016/10/12 2016.
- [5] K. Kang, S. Xie, L. Huang, Y. Han, P. Y. Huang, K. F. Mak, *et al.*, "High-mobility three-atom-thick semiconducting films with wafer-scale homogeneity," *Nature*, vol. 520, pp. 656-660, 2015/04/01 2015.
- [6] K. K. H. Smithe, S. V. Suryavanshi, M. Muñoz Rojo, A. D. Tedjarati, and E. Pop, "Low Variability in Synthetic Monolayer MoS₂ Devices," *ACS Nano*, vol. 11, pp. 8456-8463, 2017/08/22 2017.
- [7] C. Min, T. Kamins, P. V. Voorde, C. Diaz, and W. Greene, "0.18- μ m fully-depleted silicon-on-insulator MOSFET's," *IEEE Electron Device Letters*, vol. 18, pp. 251-253, 1997.
- [8] A. P. Jacob, R. Xie, M. G. Sung, L. Liebmann, R. T. P. Lee, and B. Taylor, "Scaling Challenges for Advanced CMOS Devices," *International Journal of High Speed Electronics and Systems*, vol. 26, p. 1740001, 2017.
- [9] S. Das, A. Sebastian, E. Pop, C. J. McClellan, A. D. Franklin, T. Grasser, *et al.*, "Transistors based on two-dimensional materials for future integrated circuits," *Nature Electronics*, vol. 4, pp. 786-799, 2021/11/01 2021.
- [10] R. Pendurthi, D. Jayachandran, A. Kozhakhmetov, N. Trainor, J. A. Robinson, J. M. Redwing, *et al.*, "Heterogeneous Integration of Atomically Thin Semiconductors for Non-von Neumann CMOS," *Small*, vol. 18, p. 2202590, 2022.
- [11] L. Wang, Y. Wang, J. I. Wong, T. Palacios, J. Kong, and H. Y. Yang, "Functionalized MoS₂ Nanosheet-Based Field-Effect Biosensor for Label-Free Sensitive Detection of Cancer Marker Proteins in Solution," *Small*, vol. 10, pp. 1101-1105, 2014.
- [12] A. Shokri and N. Salami, "Gas sensor based on MoS₂ monolayer," *Sensors and Actuators B: Chemical*, vol. 236, pp. 378-385, 2016.
- [13] J. Chen, L. Li, W. Ran, D. Chen, L. Wang, and G. Shen, "An intelligent MXene/MoS₂ acoustic sensor with high accuracy for mechano-acoustic recognition," *Nano Research*, vol. 16, pp. 3180-3187, 2023/02/01 2023.
- [14] A. Daus, M. Jaikissoon, A. I. Khan, A. Kumar, R. W. Grady, K. C. Saraswat, *et al.*, "Fast-Response Flexible Temperature Sensors with Atomically Thin Molybdenum Disulfide," *Nano Letters*, vol. 22, pp. 6135-6140, 2022/08/10 2022.
- [15] A. Dodda, D. Jayachandran, A. Pannone, N. Trainor, S. P. Stepanoff, M. A. Steves, *et al.*, "Active pixel sensor matrix based on monolayer MoS₂ phototransistor array," *Nature Materials*, vol. 21, pp. 1379-1387, 2022/12/01 2022.

REVIEWERS' COMMENTS

Reviewer #1 (Remarks to the Author):

The authors have well revised the manuscript and answered the reviewers' comments. Thus, it can be accepted.

Reviewer #2 (Remarks to the Author):

The authors have addressed all the major concerns, including paths for improving some of the performance of the devices that were not optimal or at the level of state of the art.

Given the novelty of the paper and the changes proposed, I recommend proceeding with the publication of the paper as it is.

Reviewer #1 (Remarks to the Author):

The authors have well revised the manuscript and answered the reviewers' comments. Thus, it can be accepted.

We express our gratitude to the reviewer for appreciating our response and recommending the acceptance of our manuscript.

Reviewer #2 (Remarks to the Author):

The authors have addressed all the major concerns, including paths for improving some of the performance of the devices that were not optimal or at the level of state of the art. Given the novelty of the paper and the changes proposed, I recommend proceeding with the publication of the paper as it is.

We express our gratitude to the reviewer for appreciating our response and recommending the acceptance of our manuscript.